# GATA3 and MDM2 are synthetic lethal in estrogen receptor-positive breast cancers

Gaia Bianco[1], Mairene Coto-Llerena[1,2], John Gallon [1], Venkatesh Kancherla[2], Stephanie Taha-Mehlitz[1], Mattia Marinucci [1], Martina Konantz [3], Sumana Srivatsa [4], Hesam Montazeri[2,5], Federica Panebianco[1], Vijaya G. Tirunagaru[6], Marta De Menna[7], Viola Paradiso[1,2], Caner Ercan [2], Ahmed Dahmani[8], Elodie Montaudon[8], Niko Beerenwinkel [4], Marianna Kruithof-de Julio [7], Luigi M. Terracciano[2,9,10], Claudia Lengerke [3], Rinath M. Jeselsohn [11], Robert C. Doebele [6], François-Clément Bidard [12], Elisabetta Marangoni [8], Charlotte K. Y. Ng [2,13,14,15 ✉] & Salvatore Piscuoglio [1,2,15 ✉]

Synthetic lethal interactions, where the simultaneous but not individual inactivation of two genes is lethal to the cell, have been successfully exploited to treat cancer. *GATA3* is frequently mutated in estrogen receptor (ER)-positive breast cancers and its deficiency defines a subset of patients with poor response to hormonal therapy and poor prognosis. However, GATA3 is not yet targetable. Here we show that *GATA3* and *MDM2* are synthetically lethal in ER-positive breast cancer. Depletion and pharmacological inhibition of MDM2 significantly impaired tumor growth in *GATA3*-deficient models in vitro, in vivo and in patient-derived organoids/xenograft (PDOs/PDX) harboring *GATA3* somatic mutations. The synthetic lethality requires p53 and acts via the PI3K/Akt/mTOR pathway. Our results present MDM2 as a therapeutic target in the substantial cohort of ER-positive, *GATA3*-mutant breast cancer patients. With MDM2 inhibitors widely available, our findings can be rapidly translated into clinical trials to evaluate in-patient efficacy.

[1] Visceral Surgery and Precision Medicine Research Laboratory, Department of Biomedicine, University of Basel, Basel, Switzerland. [2] Institute of Medical Genetics and Pathology, University Hospital Basel, Basel, Switzerland. [3] Department of Biomedicine, University of Basel and University Hospital Basel, Basel, Switzerland. [4] Department of Biosystems Science and Engineering, ETH Zurich, Basel, Switzerland. [5] Department of Bioinformatics, Institute of Biochemistry and Biophysics, University of Tehran, Tehran, Iran. [6] Rain Therapeutics Inc, Newark, CA, USA. [7] Department of Biomedical Research, Urology Group, University of Bern, Bern, Switzerland. [8] Laboratory of Preclinical Investigation, Department of Translational Research, Institut Curie Research Center, Paris, France. [9] Department of Pathology, Humanitas Clinical and Research Center, IRCCS, Milan, Italy. [10] Department of Biomedical Sciences, Humanitas University, Milan, Italy. [11] Division of Women's Cancers, Dana-Farber Cancer Institute, Harvard Medical School, Boston, MA, USA. [12] Department of Medical Oncology, Institut Curie, Saint Cloud, France. [13] Department for BioMedical Research (DBMR), University of Bern, Bern, Switzerland. [14] SIB Swiss Institute of Bioinformatics, Lausanne, Switzerland. [15] These authors jointly supervised this work: Charlotte K. Y. Ng, Salvatore Piscuoglio. ✉email: charlotte.ng@dbmr.unibe.ch; s.piscuoglio@unibas.ch

GATA3 is mutated in 12–18% of primary and metastatic estrogen receptor (ER)-positive breast cancers, with predominantly frameshift mutations and mutations affecting splice sites[1–4]. It is the most highly expressed transcription factor in the mammary epithelium[5] and has key functions in mammary epithelial cell differentiation[5]. In breast cancer, GATA3 suppresses epithelial-to-mesenchymal transition[6] and acts as a pioneer transcription factor by recruiting other cofactors, such as ERα and FOXA1[7,8]. Its expression level is strongly associated with ERα expression and is diagnostic of the luminal A and luminal B subtypes. Indeed, GATA3 loss of expression has also been strongly linked to poor response to hormonal therapy and poor prognosis[9–12]. Therefore, targeting GATA3 alterations may provide a specific and tailored treatment for a subclass of patients associated with a worse prognosis and relapse.

Synthetic lethality refers to the interaction between genetic events in two genes whereby the inactivation of either gene results in a viable phenotype, while their combined inactivation is lethal[13]. It has helped extend precision oncology to targeting genes with loss-of-function alterations by disrupting the genetic interactors of the mutated gene. One such example is the use of poly(ADP-ribose) polymerase (PARP) inhibition in cancers with deficiencies in homologous recombination[14]. Recent developments in large-scale perturbation screens have enabled the comprehensive screening of genetic interactions[15] and the systematic analysis of these screens has led to the discovery of further synthetic lethal targets in cancer[16,17]. In this study, using our recently developed SLIdR (Synthetic Lethal Identification in R) algorithm[18], we systematically interrogate the project DRIVE RNAi screen[15] and identify MDM2 as a synthetic lethal interactor of GATA3 in ER-positive breast cancer. We show that inhibition of MDM2 is synthetically lethal in GATA3-mutant and GATA3-depleted breast cancer cells. Our findings establish a new approach for targeting GATA3 deficiency in ER-positive breast cancer by pharmacological inhibition of MDM2 using selective small molecules which are currently being evaluated in clinical trials[19].

## Results

### GATA3 and MDM2 are synthetic lethal in ER-positive breast cancer

Most GATA3 mutations in ER+ breast cancer introduce frameshifts or alternative splicing resulting in protein truncation or extension[20], with 89% of them predicted to be driver mutations (Fig. 1a). To identify synthetically lethal vulnerabilities of GATA3 in breast cancer, we analyzed the breast cancer cell line ($n = 22$, Supplementary Data 1) data from the large-scale, deep RNAi screen project DRIVE[15] using our recently developed SLIdR algorithm[18]. SLIdR uses rank-based statistical tests to compare the viability scores for each gene knock-down between the GATA3-mutant and GATA3-wild type cell lines (Fig. 1b) and identified 13 synthetic lethal partners of GATA3 (FDR < 0.05, Supplementary Data 2). We interrogated the candidates for well-developed drug targets and identified MDM2 as the top druggable gene whose knock-down significantly reduced cell viability in the two GATA3-mutant breast cancer cell lines (MCF-7 and KPL-1, Fig. 1c, d). MDM2 encodes an E3 ubiquitin ligase that inhibits the tumor suppressor p53-mediated transcriptional activation[21] and is frequently amplified and overexpressed in human cancers, including breast[22].

We first sought to validate the predicted synthetic lethality between GATA3 and MDM2 in the ER-positive breast cancer cell line MCF-7, one of the two GATA3-mutant cell lines used in the RNAi screen[15]. MCF-7 harbors the GATA3 frameshift mutation p.D335Gfs[23], a truncating mutation recurrently observed in breast cancer patients[1,4] and that has been reported to have both

loss and gain-of-function effects, and specifically acts as a dominant-negative mutant with lower DNA binding affinity but increased half-life[24]. Using a siRNA approach, we confirmed that silencing MDM2 significantly reduced cell proliferation in MCF-7 cells (Fig. 1e and Supplementary Fig. 1a). MDM2 siRNA titration analysis showed that the vulnerability induced by MDM2 inhibition in MCF-7 was dose-dependent and that 50% reduction in MDM2 expression is sufficient to inhibit proliferation in the presence of GATA3 mutation (Supplementary Fig. 1b, c).

To confirm that the effect of MDM2 silencing is unequivocally related to GATA3 loss of function and to exclude any gain-of-function effects of the GATA3 mutation, we assessed the changes in cell proliferation upon single- and dual-silencing of GATA3 and MDM2 using siRNA in two ER-positive GATA3-wild type breast cancer cell lines, the luminal A (ER+/HER2−) MDA-MB134 and the luminal B (ER+/HER2+) BT-474 (Supplementary Fig. 1d, e). Consistent with the oncosuppressor role of GATA3 in breast cancer[25,26], GATA3-silencing led to a significant increase in cell proliferation in both BT-474 and MDA-MB134 (Fig. 1f, g). By contrast, dual-silencing of GATA3 and MDM2 significantly reduced cell proliferation compared to cells transfected with control siRNA, GATA3 siRNA, or MDM2 siRNA alone (Fig. 1f, g).

To determine if MDM2 silencing was merely inhibiting cell growth or was actively inducing cell death, we assessed apoptosis using Annexin V and propidium iodide co-staining followed by flow cytometry analysis. We observed that MDM2 silencing significantly induced apoptosis in MCF-7 cells in a dose-dependent manner (Fig. 1h and Supplementary Fig. 1f). Similarly, dual-GATA3/MDM2 silencing in BT-474 and MDA-MB134 cells led to a 15–20% higher proportion of apoptotic cells than the silencing of the two genes individually (Fig. 1h), indicating that dual inhibition induced increased apoptosis.

Our results provide evidence that MDM2 is a selected vulnerability in breast cancer with GATA3-mutation and/or loss of GATA3.

### GATA3 status determines response to MDM2 inhibitors in vitro

The selected vulnerability of MDM2 in GATA3-deficient ER-positive breast cancers presents MDM2 as an attractive therapeutic target in this patient cohort. To test whether the apoptotic effects of MDM2 inhibition could be achieved using an MDM2 antagonist, we treated the breast cancer cell lines with idasanutlin (RG7388, Supplementary Fig. 2a)[27,28]. In the GATA3-mutant MCF-7 cells, idasanutlin induced cell growth arrest and apoptosis in a dose-dependent manner (Fig. 2a, b). To assess whether idasanutlin was inducing the canonical apoptotic cascade, we assessed the expression of Bax and Bcl-2, together with the canonical markers of apoptosis PARP and cleaved PARP, by immunoblot at 6, 12, and 24 h post-treatment. Idasanutlin induced an early up-regulation of MDM2 protein[29], together with the up- and down-regulation of pro- and anti-apoptotic proteins, respectively (Fig. 2c), leading to the activation of the apoptotic cascade. To demonstrate that MCF-7 sensitivity was indeed due to the presence of mutant GATA3, we rescued GATA3 wild-type expression in MCF-7 cells (Fig. 2d and Supplementary Fig. 2b). Notably, MCF-7 cells overexpressing wild-type GATA3 proliferated significantly less compared to control cells (Fig. 2d). More importantly, rescuing wild-type GATA3 desensitized cells to idasanutlin (Fig. 2d). Indeed, and contrary to control cells, idasanutlin did not significantly affect cell viability in MCF-7 cells where GATA3 wild-type was overexpressed (Fig. 2d).

To determine whether GATA3 expression levels would modulate response to idasanutlin, we assessed the effect of treatment on GATA3-silenced BT-474 and MDA-MB134 cells.

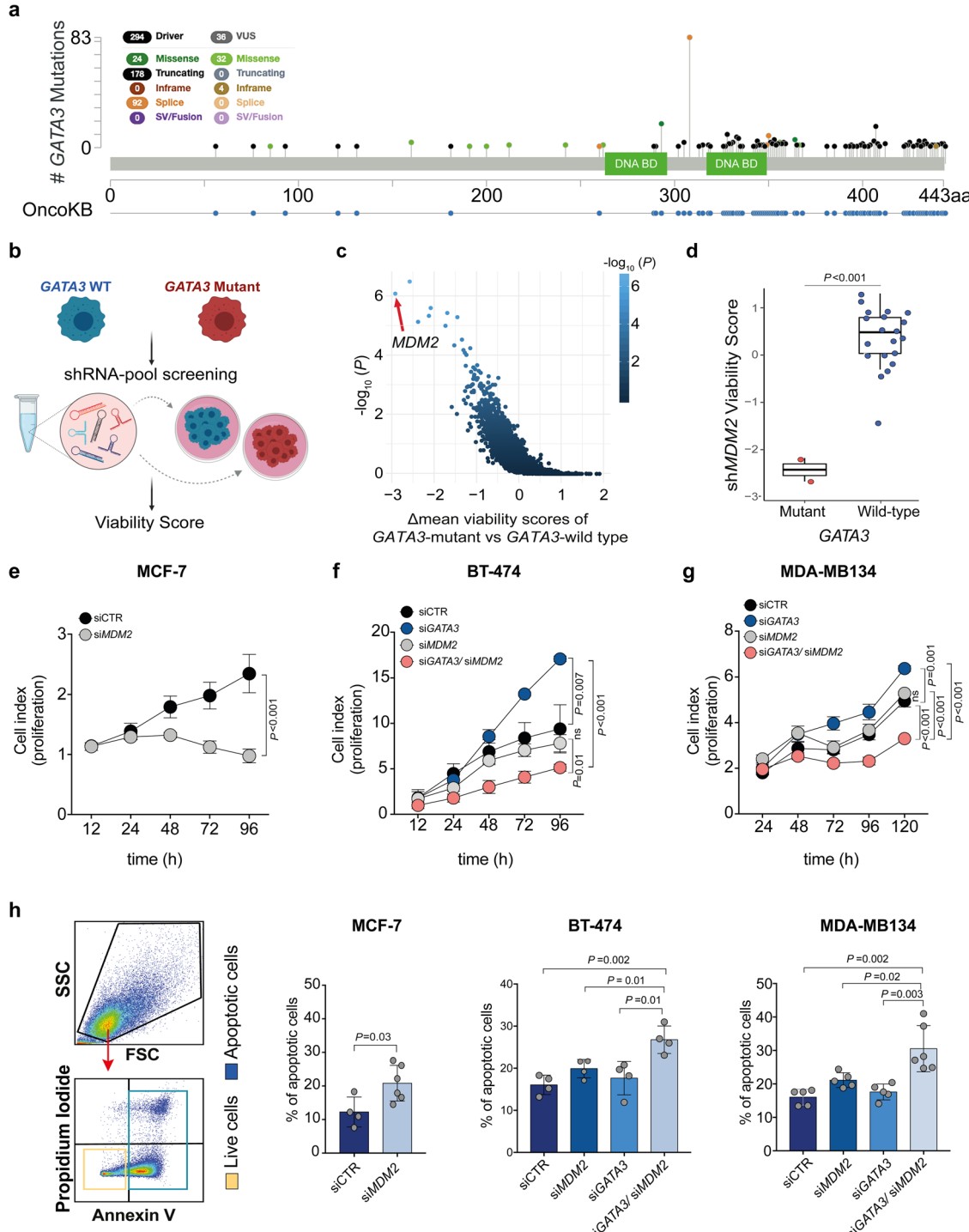

**Fig. 1 GATA3 and MDM2 are synthetic lethal in ER-positive breast cancer. a** Lollipop plot depicting GATA3 somatic mutations and OncoKB[96] annotation in ER+ breast cancer derived from the TCGA PanCancer Atlas[1] and the METABRIC datasets[4]. **b** Schematic representation of the project DRIVE shRNA screen data used to identify synthetic lethal interactors of GATA3. **c** SLIdR-derived statistical significance (-log₁₀(P)) plotted against the difference in the mean viability scores between GATA3-mutant and GATA3-wild type breast cancer cell lines for each gene knocked down in the shRNA screen. The middle lines of the boxplots indicate medians. Box limits are first and third quartiles. The whiskers extend to the range. **d** Viability scores of MDM2 knock-down in GATA3-mutant and GATA3-wild type cell lines. **e–g** Proliferation kinetics of **e** GATA3-mutant MCF-7 transfected with siRNA targeting MDM2 or control (see also Supplementary Fig. 1a–c), **f** GATA3-wild type BT-474, **g** GATA3-wild type MDA-MB134 transfected with siRNA targeting GATA3, MDM2, GATA3/ MDM2, or control (see also Supplementary Fig. 1d, e). **h** Apoptosis assay using Annexin V and propidium iodide co-staining. From left: gating strategy to define apoptotic (blue) and live (yellow) cells; percentage of apoptotic and live cells upon MDM2 silencing in MCF-7 (see also Supplementary Fig. 1f) upon silencing of GATA3 and MDM2 alone or in combination in BT-474 and MDA-MB134. Data are mean ± s.d. n ≥ 3 biologically independent replicates. Statistical significance was determined for **e–g** by multiple t test and for **h** by two-tailed unpaired Student's t test. **b** was created with BioRender.com.

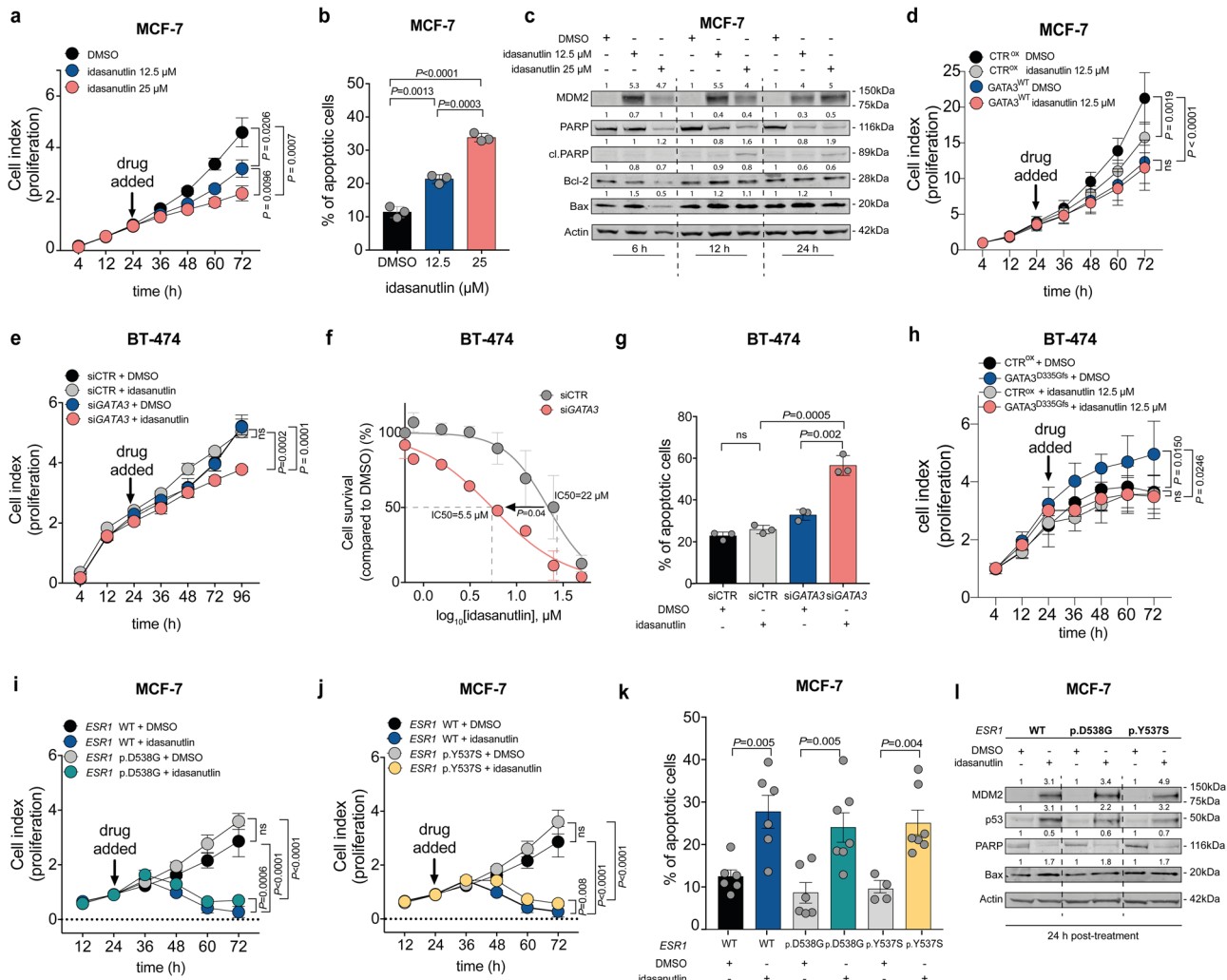

**Fig. 2 GATA3 status determines response to MDM2 inhibitor in vitro. a**, **d**, **e**, **h**, **i**, **j** Proliferation kinetics of **a** GATA3-mutant MCF-7 under increasing dosage of idasanutlin, **d** control and GATA3-WT rescued MCF-7 upon 12.5 μM idasanutlin treatment, **e** BT-474 upon GATA3 silencing and/or treatment with 12.5 μM idasanutlin, **h** BT-474 upon GATA3 p.D335Gfs overexpression and/or treatment with 12.5 μM idasanutlin, **i**, **j** GATA3-mutant MCF-7 carrying a wild-type ESR1 or mutant ESR1 (p.D538G/p.Y537S) upon treatment with 12.5 μM idasanutlin. **b**, **g**, **k** Apoptosis assay using Annexin V and propidium iodide co-staining **b** upon the increasing dosage of idasanutlin in MCF-7, **g** upon GATA3 silencing and/or treatment with 12.5 μM idasanutlin in BT-474, **k** upon treatment of 12.5 μM idasanutlin in MCF-7 carrying a wild-type ESR1 or mutant ESR1 (p.D538G/p.Y537S). **c**, **l** Immunoblot showing pro- and anti-apoptotic proteins **c** at 6, 12 and 24 h post-treatment with DMSO, 12.5 μM and 25 μM idasanutlin in MCF-7, **l** at 24 h post-treatment with DMSO or 12.5 μM idasanutlin in MCF-7 carrying wild-type or mutant ESR1 (p.D538G/p.Y537S). For all the western blots, quantification is relative to the loading control (actin) and normalized to the corresponding DMSO control. **f** Log-dose response curve of idasanutlin in BT-474 transfected with GATA3 siRNA or control siRNA (see also Supplementary Fig. 2d). Data are mean ± s.d. n ≥ 3 biologically independent experiments. Statistical significance was determined for **a**, **d**, **e**, **h**, **i**, **j** by multiple t test and for **b**, **f**, **g**, **k** by two-tailed unpaired Student's t test.

We observed that while idasanutlin treatment had no or little effect on the proliferation of the control cells, it significantly reduced cell proliferation upon GATA3 silencing (Fig. 2e and Supplementary Fig. 2c). In fact, both cell lines showed that GATA3 silencing substantially reduced the IC50 for idasanutlin (Fig. 2f and Supplementary Fig. 2d). Flow cytometry and immunoblot further demonstrated that idasanutlin treatment induced apoptosis in both BT-474 and MDA-MB134 upon GATA3 silencing but not in control cells (Fig. 2g and Supplementary Fig. 2e–g). To rule out the possibility that the differential sensitivity was derived from different genetic backgrounds of breast cancer cell lines, we asked whether overexpressing the GATA3 p.D335Gfs mutation in GATA3-wild-type cells would induce hypersensitivity against MDM2 inhibition (Supplementary Fig. 2h). Notably, BT474 cells overexpressing GATA3 p.D335Gfs mutation significantly proliferated more

compared to control cells (Fig. 2h). Similar to the phenotype induced by GATA3 gene silencing, idasanutlin significantly reduced cell proliferation in GATA3 p.D335Gfs-overexpressing cells but not in control cells (Fig. 2h). These data further support our hypothesis that the GATA3 p.D335Gfs mutant background confers synthetic lethality with MDM2 inhibition in ER-positive breast cancer cells.

Acquired resistance to endocrine therapy is often associated with ESR1 activating mutations[30] or fusion genes[31]. We hypothesized that MDM2 inhibition may represent an alternative therapeutic strategy in endocrine therapy-resistant breast cancers harboring GATA3 mutations. To test this hypothesis, we treated two derivative endocrine-resistant GATA3-mutant MCF-7 cell lines with knock-in ESR1 p.D538G or p.Y537S activating mutations[32,33] with idasanutlin. Both ESR1-mutant cells have previously been shown to exhibit estradiol(E2)-independent

growth and resistance to fulvestrant and tamoxifen[33,34]. We observed that idasanutlin stopped cell proliferation in both mutant cell lines (Fig. 2i, j). Idasanutlin also induced apoptosis and up- and down-regulated pro- and anti-apoptotic proteins, respectively (Fig. 2k, l).

Taken together, our results demonstrate that *GATA3* p.D335Gfs mutant background and *GATA3* loss of expression sensitizes cells to pharmacological inhibition of MDM2 in vitro.

**GATA3 expression determines response to MDM2 inhibitor in vivo.** To ascertain whether *GATA3* expression levels would also modulate response to idasanutlin in vivo, we performed xenotransplantation into zebrafish embryos. As a cancer model system, human cancer xenografts in zebrafish recapitulate the response to anticancer therapies of mammalian models[35,36]. Adding idasanutlin directly to the fish water is toxic to the zebrafish. Given that idasanutlin-induced apoptosis is delayed[37], and that intermittent dosing schedules (once or twice a week) of idasanutlin induce a reduction in mean tumor volume compared with continuous dosing[37], we circumvented fish toxicity by pre-treating *GATA3*-silenced and control BT-474 cells with idasanutlin (25 µM) or vehicle (DMSO) for 24 h, and 48 h post-siRNA transfection, followed by wash-out. Twenty-four hours post-treatment, we labeled the cells with a red fluorescent cell tracker, injected them into the yolk sac of zebrafish embryos and screened embryos for tumor cell engraftment after four days (Fig. 3a)[38].

We observed that *GATA3*-silenced cells injected into fish were more sensitive to idasanutlin than the control (42 vs 61%, Fig. 3b). More importantly, idasanutlin reduced tumor formation in the context of *GATA3*-silencing (42 vs 65% treated with DMSO) but not in control (61 vs 56% treated with DMSO, Fig. 3b). Tumors derived from *GATA3*-silenced, idasanutlin-treated cells, were very small, largely consisting of small clusters of tumor cells, compared to the larger solid tumor masses derived from *GATA3*-silenced cells without idasanutlin (Fig. 3c). To assess cell proliferation, we quantified the percentage of tumor cells present in the fish by performing FACS analysis of the fluorescence-labeled tumor cells in whole fish extracts. Consistent with the results from the tumor formation assay, idasanutlin treatment was only effective in reducing the overall percentage of tumor cells in fish injected with *GATA3*-silenced cells (purple vs DMSO-treated in blue) but not in fish injected with control (yellow vs DMSO-treated in black, Fig. 3d), indicating that *GATA3* expression level modulates sensitivity to MDM2 inhibition in vivo.

The zebrafish xenograft model provides insights into the tumorigenic and proliferative capability of cancer cells. However, to assess apoptosis and to quantify tumor growth, we employed the chicken chorioallantoic membrane (CAM), a densely vascularized extraembryonic tissue, as a second in vivo model[39,40]. Similar to the zebrafish assay, we treated *GATA3*-silenced and control BT-474 cells with idasanutlin (25 µM) or vehicle (DMSO) for 24 h. We then inoculated the cells into the CAMs and screened the eggs for tumor formation four days later (Fig. 3e). In accordance with our results in the zebrafish model, idasanutlin treatment reduced the volume of tumors formed by *GATA3*-silenced cells (purple vs DMSO-treated in blue) but not in control cells (yellow vs DMSO-treated in black, Fig. 3f, g), suggesting that *GATA3* expression modulates response to MDM2 inhibitors in the CAM model as well. Notably, tumors derived from *GATA3*-silenced cells were significantly larger than the control counterpart (Fig. 3g). We then evaluated apoptosis induction by staining tumor sections with the apoptotic marker cleaved caspase 3. Notably, only *GATA3*-silenced idasanutlin-treated tumors showed a strong positive signal for cleaved caspase 3, as well as morphological features of apoptosis (e.g. nuclear

fragmentation, hypereosinophilic cytoplasm, "apoptotic bodies," Fig. 3h and Supplementary Fig. 3), demonstrating that idasanutlin induces apoptosis in the context of *GATA3* silencing in vivo.

Taken together, our results show that *GATA3* expression modulates response to idasanutlin in two independent in vivo models.

**The synthetic lethality between GATA3 and MDM2 is TP53 dependent.** MDM2 plays a central role in the regulation of p53 and they regulate each other in a complex regulatory feedback loop[41] (Fig. 4a). We analyzed the frequencies of *GATA3* and *TP53* mutations in ER-positive breast cancer[1,4] and observed that they are mutually exclusive (Fig. 4b). We, therefore, hypothesized that the synthetic lethal effects between *GATA3* and *MDM2* may be p53 dependent. To test this hypothesis, we assessed cell growth and apoptosis upon single- and dual-silencing of *GATA3* and *MDM2* in the ER-positive, *GATA3*-wild-type, *TP53*-mutant (p.L194F) T-47D breast cancer cell line (Supplementary Fig. 4a). Consistent with the mutual exclusivity of *GATA3* and *TP53* mutations, *GATA3* silencing in a *TP53*-mutant context resulted in a strong reduction of cell viability and induction of apoptosis (Fig. 4c, d). Contrary to the results obtained in cells with functional p53, *GATA3/MDM2* dual silencing did not show a synthetic lethal effect (Fig. 4c, d). If the synthetic lethal interaction between *GATA3* and *MDM2* is *TP53*-dependent, one should expect that silencing *TP53* should partially revert the phenotype. Therefore, we silenced *MDM2* alone or in combination with *TP53* in the *GATA3*-mutant MCF-7 cell line (Supplementary Fig. 4b). As expected, *TP53* silencing partially rescued the effect induced by *MDM2* knockdown (Fig. 4e, f) as well as of idasanutlin treatment (Fig. 4g, h and Supplementary Fig. 4c, d) on cell growth and apoptosis, demonstrating the p53 dependency of the synthetic lethal interaction.

**GATA3 mutational status predicts response to MDM2 inhibitors in ER-positive breast cancer patient-derived organoids (PDOs) and patient-derived xenograft (PDX).** As patient-derived organoids (PDOs) have been shown to retain the molecular features of the original tumors and to better resemble tumor heterogeneity than traditional two-dimensional cell culture methods derived from single-cell clones, they are frequently used as ex vivo preclinical models for drug response prediction[42–45]. Indeed, drug sensitivity of PDOs has been shown to mirror the patient's response in the clinic[46,47]. We therefore tested our findings in organoids derived from three ER-positive invasive ductal breast carcinoma patients, a primary tumor, a bone metastasis harboring *GATA3* frameshift mutations (primary: p.H433fs and metastasis: p.S410fs, Supplementary Fig. 5a) and one bone metastasis carrying wild-type *GATA3* (Fig. 5a). Importantly, all tumor specimens were carrying a wild-type *TP53* gene, retained ERα and GATA3 expression (Fig. 5b and Supplementary Fig. 5b), and have been previously shown to be estrogen responsive in vivo[48]. Both p.H433fs and p.S410fs *GATA3* mutations result in elongated protein isoforms (51 and 55 kD, respectively), with the primary tumor (HBCx-169) showing no GATA3 wild type protein expression (Supplementary Fig. 5c). Gene expression analysis showed no such clear segregation between *GATA3* mutants and wild-type PDXs based on *MDM2* expression (Supplementary Fig. 5d). In accordance with the oncosuppressor role of GATA3 and our previously generated data, PDOs derived from the *GATA3*-mutant primary breast cancer significantly proliferated more compared to *GATA3* wild-type PDOs (Fig. 5c). In accordance with the results generated in vitro and in vivo, *GATA3*-mutant PDOs showed a significant decrease in IC50 for idasanutlin (Fig. 5d and Supplementary Fig. 5e). Furthermore, viability assay revealed that both *GATA3*-

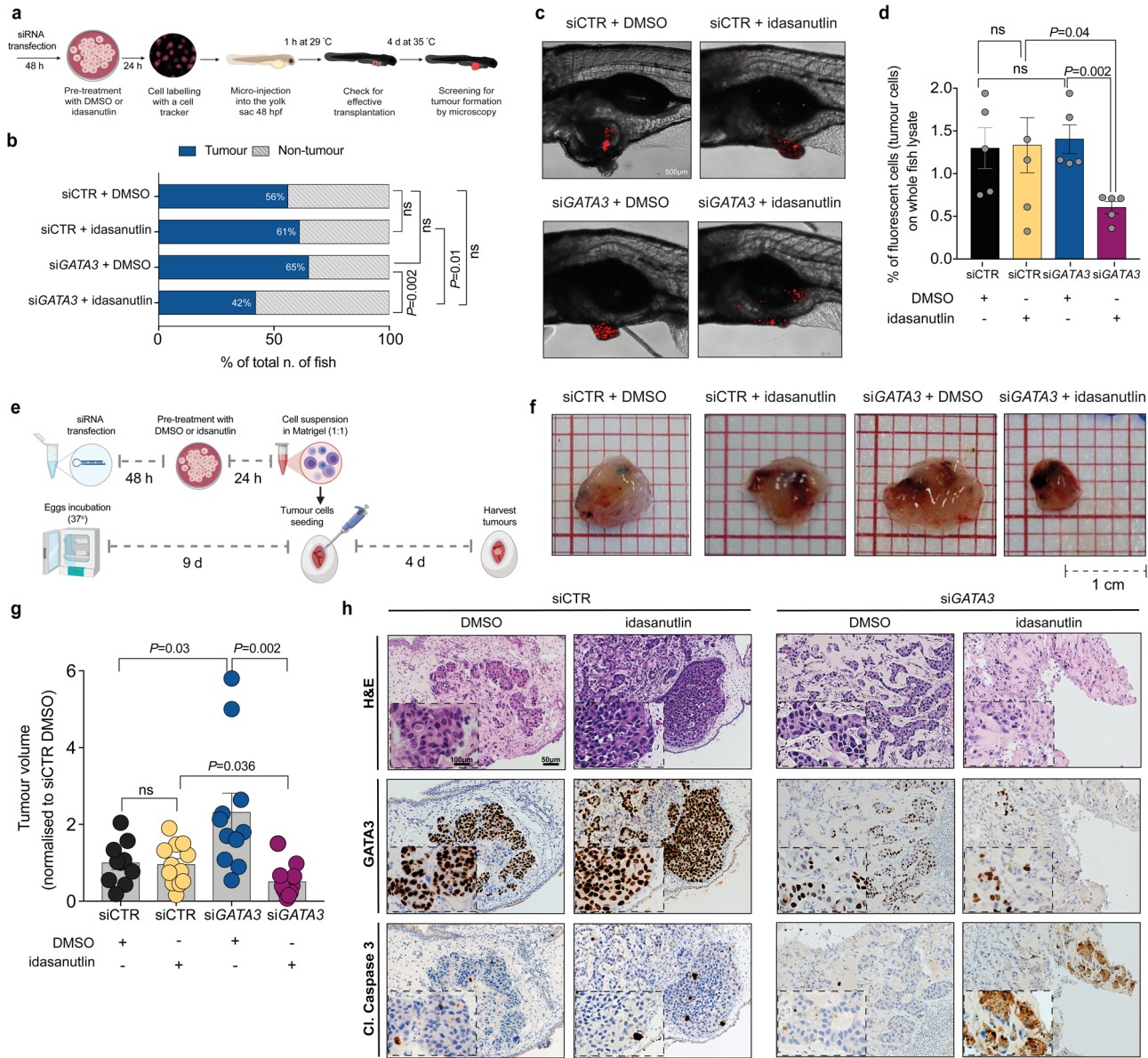

**Fig. 3 GATA3 expression determines response to MDM2 inhibitor in vivo. a** Schematic representation of the zebrafish xenotransplantation assay. **b** Barplot shows the percentages of fish that harbored or did not harbor tumors upon transplantation with *GATA3*-silenced or control BT-474 cells pre-treated with idasanutlin or DMSO. In total, 70–100 embryos per group were analyzed over two independent experiments. **c** Representative confocal images of tumor formation in zebrafish injected with fluorescent tracker-labeled BT-474 cells with *GATA3* siRNA or control siRNA, pretreated with idasanutlin or DMSO. **d** FACS analysis showing the percentage of red-tracker labeled tumor cells extracted from the embryos. Error bars represent, in total, three replicates performed over two independent experiments. Each replicate represents the pooled lysate of 20-30 fish for each condition. **e** Schematic illustration of the CAM assay. **f** Photographs of *GATA3*-silenced or control BT-474 cells pre-treated with DMSO or idasanutlin implanted in CAMs and grown for 4 days post-implantation. **g** Volume of tumors derived from the CAM experiment ($n \geq 10$ tumors over three independent experiments). Values are normalized to the mean of siCTR DMSO. **h** Representative micrographs of BT-474 tumors extracted 4 days post-implantation. Tumoral cells (upper) were immunostained with GATA3 (middle) and the apoptotic marker cleaved caspase 3 (lower) in the different treatment conditions (see also Supplementary Fig. 3). Data are mean ± SEM $n \geq 4$ biologically independent experiments. Scale bars: **c** 500 μm, **f** 1 cm and **h** 50 and 100 μm. Statistical significance was determined for **b** by two-sided Fisher's Exact test and for **d**, **g** by two-tailed unpaired Student's *t* test. **a**, **e** were created with BioRender.com.

mutant PDOs treated with idasanutlin significantly proliferated less (~50% for the primary and ~25% for the bone metastasis) compared to the wild-type PDOs (Fig. 5e, f and Supplementary Fig. 5f). In particular, upon treatment with 1.5 μM idasanutlin *GATA3* wild-type PDOs were still 100% alive compared to their DMSO control counterparts while only 40% (primary) and 60% (metastasis) *GATA3*-mutant PDOs survived (Fig. 5e, f and Supplementary Fig. 5f).

To confirm that the effect of idasanutlin was indeed due to MDM2 inhibition rather than off-target effects, we tested additional clinical-grade MDM2 inhibitors, specifically MI-773 (SAR405838)[49] and RAIN-32 (milademetan[50], Supplementary Fig. 5g). Similar to idasanutlin, both inhibitors showed significantly higher efficacy in *GATA3*-mutant compared to wild-type PDOs (Fig. 5g and Supplementary Fig. 5h). We further tested the antitumor effect of MDM2 inhibition over time in a PDX model of an ER-positive breast

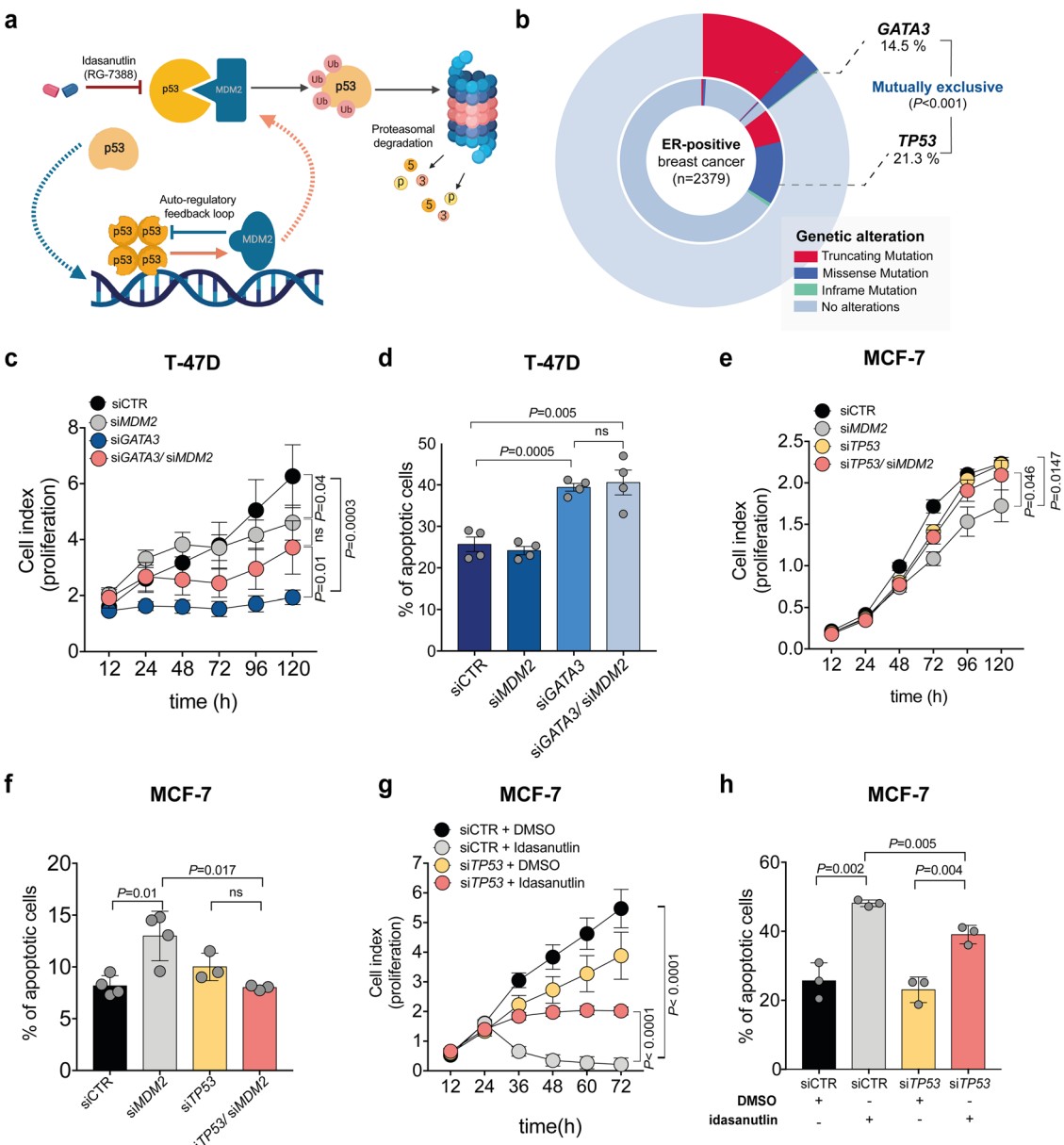

**Fig. 4 The synthetic lethality between *GATA3* and *MDM2* is *TP53* dependent. a** Schematic representation of the regulatory feedback loop between MDM2 and p53. **b** Doughnut chart showing *GATA3* and *TP53* mutations in ER-positive breast cancer. Mutational data were derived from the TCGA PanCancer Atlas[1] and the METABRIC datasets[4]. **c** Proliferation kinetics of *TP53*-mutant T-47D transfected with siRNA targeting *GATA3, MDM2, GATA3/ MDM2*, or control (see also Supplementary Fig. 4a). **d** Percentage of apoptotic cells upon silencing of *GATA3* and *MDM2* alone or in combination in T-47D. **e** Proliferation kinetics of MCF-7 transfected with siRNA targeting *TP53, MDM2, TP53/MDM2*, or control (see also Supplementary Fig. 4B). **f** Percentage of apoptotic cells upon silencing of *TP53* or *MDM2* alone or in combination in MCF-7. **g** Proliferation kinetics of MCF-7 upon *TP53* silencing and/or treatment with 12.5 μM idasanutlin (see also Supplementary Fig. 4c). **h** Percentage of apoptotic cells upon silencing of *TP53* and/or treatment with 12.5 μM idasanutlin (see also Supplementary Fig. 4d). Data are mean ± s.d. n ≥ 3 biologically independent replicates. Statistical significance was determined for **b** by one-sided Fisher's Exact test, for **c**, **e**, **g** by multiple *t* test, and for **d**, **f**, **h** by two-tailed unpaired Student's *t* test. **a** was created with BioRender.com.

cancer harboring the *GATA3* p.D335Gfs mutation (Fig. 5h). PDX models have been shown to be the most representative preclinical models for drug development in several cancer types[51,52], as these models maintain the biological characteristics of the donor tumors, and there is a high degree of correlation between clinical response in patients and response to the same agent in PDX models generated from these patients[51,52]. NOD/SCID mice were inoculated subcutaneously with *GATA3* p.D335Gfs mutant breast tumor chunks (2–3 mm) for tumor development (Fig. 5h). When the mean tumor size reached approximately 150 mm³, PDX-inoculated mice were randomized for treatment for 29 days with one of the following:

vehicle, fulvestrant (200 mg/kg), RAIN-32 (50 mg/kg), RAIN-32 (100 mg/kg), or a combination of fulvestrant (200 mg/kg) and RAIN-32 (100 mg/kg). Notably, while PDX-inoculated mice did not respond to treatment with fulvestrant, both 50 and 100 mg/kg doses of RAIN-32 were highly effective in reducing, and almost inhibiting, tumor growth (Fig. 5i). Similarly, treatment with fulvestrant in combination with RAIN-32 significantly reduced tumor growth when compared to treatment with fulvestrant alone (Fig. 5i).

Our ex vivo and in vivo data derived from PDOs and PDX models further support the use of MDM2 inhibition in the treatment of *GATA3*-mutant ER-positive breast cancer patients, in

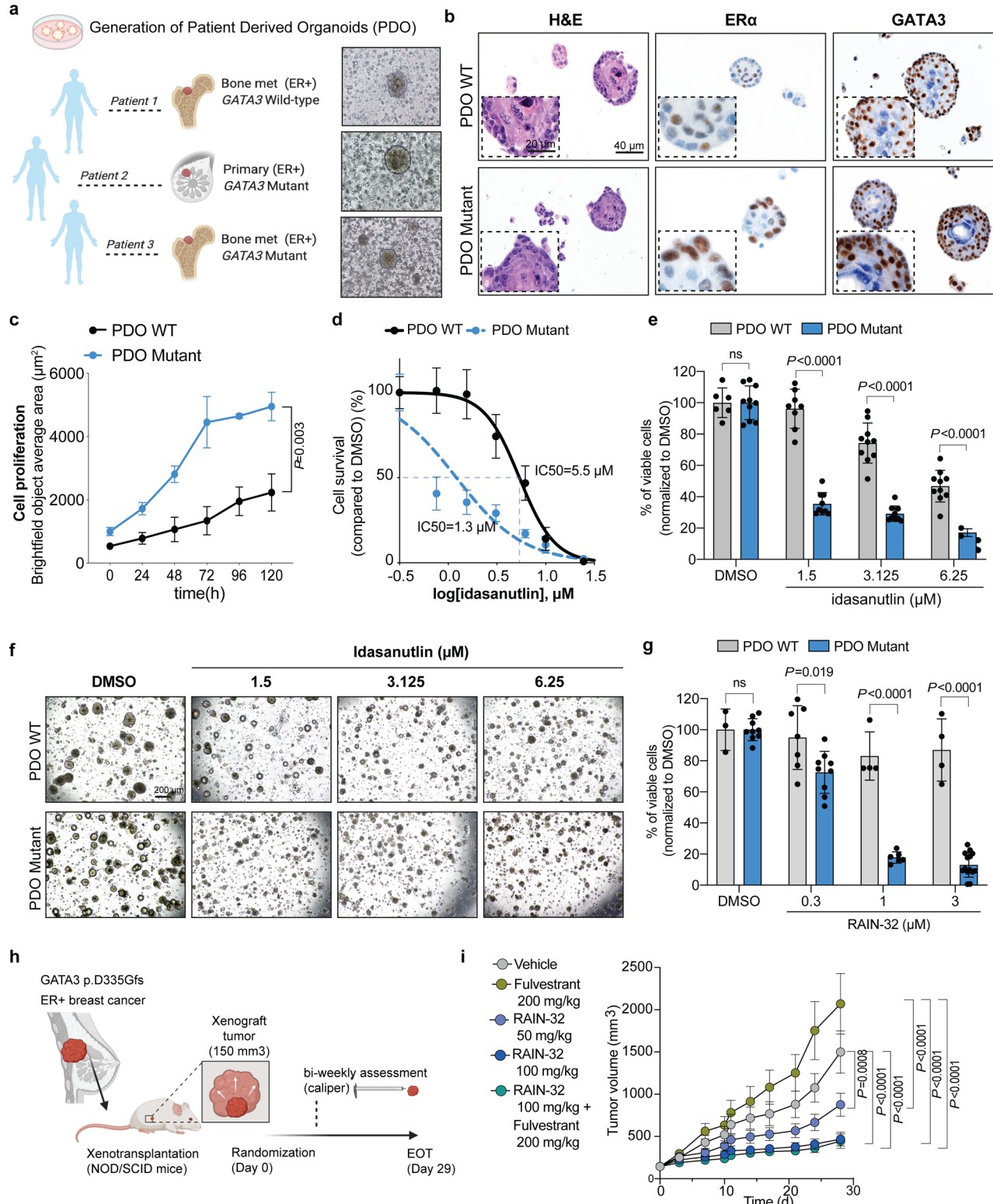

particular as a combination with first-line treatment with anti-estrogen therapy, or as an alternative for those patients developing resistance to fulvestrant.

**The synthetic lethality between *GATA3* and *MDM2* acts via the PI3K-Akt-mTOR signaling pathway.** To investigate the putative mechanisms driving the synthetic lethality, we analyzed the gene

expression changes induced by concurrent *GATA3* loss and *MDM2* silencing. RNA-sequencing analysis of the *MDM2*-silenced MCF-7 cells and dual *GATA3/MDM2*-silenced MDA-MB134 cells revealed 20 commonly dysregulated pathways (Fig. 6a). As expected, pathways related to p53 and apoptosis were significantly up-regulated in both cell lines, while many proliferation-related pathways such as *E2F* and *MYC* targets were down-regulated (Fig. 6b). Interestingly, the mTORC1 signaling

**Fig. 5 GATA3 mutations predict response to MDM2 inhibitors in ER-positive breast cancer PDOs and PDX. a** Schematic representation and representative microscopy pictures of the generation of organoids from $n = 3$ ER-positive breast cancers (see also Supplementary Fig. 5). **b** Representative micrographs of H&E, ERα, and GATA3 immuno-staining on the PDOs (see also Supplementary Fig. 5b). **c** Proliferation kinetics of GATA3-wild-type PDOs (black, patient 1) and GATA3-mutant PDOs (blue, patient 2). **d** Log-dose response curve of idasanutlin in GATA3-wild-type (IC50 = 5.4 μM) or GATA3-mutant (1.2 μM) PDOs (see also Supplementary Fig. 5e). **e** Percentage of viable cells upon treatment with different dosages of idasanutlin in GATA3-wild-type (gray) or GATA3-mutant (blue) PDOs (see also Supplementary Fig. 5f). **f** Representative micrographs of PDOs after five days of treatment with different dosages of idasanutlin. Scale bars are 20 and 40 μm for (**b**) and 200 μm for (**f**). **g** Percentage of viable cells upon treatment with different dosages of RAIN-32 in GATA3-wild-type (gray) or GATA3-mutant (blue) PDOs. **h** Schematic representation of the PDX model and drug treatment. **i** Tumor growth curve of GATA3 p.D335fs PDXs ($n = 6$–8 mice) treated for 29 days with vehicle, fulvestrant (200 mg/kg), RAIN-32 (50 and 100 mg/kg) alone, or RAIN-32 in combination with fulvestrant. Data are mean ± SD, $n \geq 3$ biologically independent replicates. Statistical significance was determined for **c, e, g, i** by multiple $t$ test and for **d** by two-tailed unpaired Student's $t$ test. **a, h** were created with BioRender.com.

pathway was among the most significantly down-regulated pathways in both cell lines. Indeed, we confirmed that MDM2 silencing in the GATA3-mutant MCF-7 cells (Supplementary Fig. 6a) reduced phospho-Akt, phospho-S6, as well as phospho-GSK3β, compared to control cells (Fig. 6c and Supplementary Fig. 6b), indicating the down-regulation of the mTOR pathway. Similarly, in BT-474 cells, dual GATA3/MDM2 silencing (Supplementary Fig. 6a) reduced levels of phospho-Akt, phospho-S6 and phospho-GSK3β and induced apoptosis (Fig. 6d and Supplementary Fig. 6c). By contrast, phospho-Akt levels were higher when only GATA3 was silenced (Fig. 6d and Supplementary Fig. 6c). Pharmacological inhibition of MDM2 in GATA3-silenced BT-474 cells also resulted in a reduction in phospho-Akt, phospho-S6, and phospho-GSK3β (Supplementary Fig. 6d). To determine whether deregulation of the mTOR signaling cascade could also be observed in vivo, we stained the tumors in our CAM model with phospho-S6 and phospho-Akt. Indeed, in tumors derived from GATA3-silenced BT-474 cells, both phospho-S6 and phospho-Akt were reduced upon treatment with idasanutlin, while in tumors derived from control cells, idasanutlin treatment did not have an effect on mTOR signaling (Fig. 6e and Supplementary Fig. 6e).

We, therefore, hypothesized that GATA3 loss may induce addiction to mTOR signaling in breast cancer cells. To functionally validate our hypothesis, we assessed the phosphorylation level of the S6 protein upon the rescue of wild-type GATA3 in MCF-7 cells (Supplementary Fig. 6f). Indeed, rescuing wild-type GATA3 resulted in a 30% reduction of phospho-S6 72 h post-transfection (Supplementary Fig. 6f). Additionally, pharmacological inhibition of MDM2 in control cells resulted in a reduction in phospho-S6 compared to DMSO, but failed to significantly alter phospho-S6 level in wild-type GATA3-rescued cells compared to their vehicle counterpart (Supplementary Fig. 6g). These results suggest that GATA3 mutations activate the mTOR signaling cascade, and that the effect of MDM2 inhibition on the mTOR pathway is dependent on the presence of a GATA3-mutant protein. In further support of our hypothesis, we observed that, in ER-positive breast cancers, genetic alterations in GATA3 are significantly mutually exclusive with those in both PI3KCA and PTEN (Supplementary Fig. S7a, b). Furthermore, differential gene expression and pathway enrichment analyses between GATA3-mutant and GATA3-wild type ER-positive breast cancers and between ER-positive breast cancers with low and high GATA3 expression levels also showed significant enrichment for the mTORC1 signaling pathway (Supplementary Fig. 7c, d).

Taken together, our results show that the synthetic lethality between GATA3 and MDM2 acts at least partially via the PI3K-Akt-mTOR signaling pathway.

## Discussion
GATA3 is mutated in 12–18% of breast cancer[1,2] with predominantly frameshift mutation resulting in protein truncation

or extension[20]. These mutations mostly act in a dominant-negative manner by impairing the wild-type function[24,53] through diverse mechanisms such as alteration of protein stability[54], aberrant nuclear localization, decrease in transcription activation[24,55], and loss of DNA binding[54], all resulting in the loss of canonical GATA3 functions and reprogramming of the transcriptional network[54]. Additionally, loss of GATA3 expression is strongly associated with failure to respond to hormonal therapy and poor prognosis[11]. Here we describe a synthetic lethal interaction between GATA3 and MDM2 in ER-positive breast cancer. In particular, we showed that, in the context of both truncating (p.D335fs) and elongating (pH433fs, pS410fs) GATA3 mutations, inhibition of MDM2 hampers cancer cell proliferation and tumor growth in vitro, in three independent in vivo models (zebrafish, CAM and canonical PDX) and in two GATA3-mutant PDOs. Re-expression of wild-type GATA3 in mutant cells rescued the effects of MDM2 inhibition and, on the contrary, forced expression of mutant GATA3 sensitized wild-type cells to idasanutlin. In the context of wild-type GATA3, the same effect was achieved by dual GATA3 and MDM2 inhibition, regardless of HER2 status. We further showed that GATA3 expression level modulates response to MDM2 inhibitors. Of note, our data suggest that MDM2 inhibitors might be efficacious on different classes of GATA3 somatic mutations, mostly frameshift truncating and elongating mutations affecting the stability or transactivation activity of the wild-type protein through diverse mechanisms[24,55]. Our results thus support MDM2 as a therapeutic target in the substantial fraction of ER-positive, GATA3-deficient breast cancer.

We showed that the synthetic lethality between GATA3 and MDM2 is p53-dependent and acts at least partially via the PI3K/Akt/mTOR pathway. It is well known that in normal conditions p53 and the PI3K/Akt/mTOR pathway co-regulate cell cycle arrest and apoptosis leading to homeostasis between cell death and survival[56,57]. Our results suggest that in breast cancer cells, GATA3 loss-of-function (via genetic alterations or other mechanisms) activates the PI3K/Akt/mTOR pathway and leads to resistance to apoptosis. In this context, MDM2 inhibition, with consequent p53 up-regulation and mTOR signaling down-regulation, pushes the cells toward cell death (Fig. 6f). In support of this model, down-regulation of GATA3 has been directly linked to Akt kinase activation in breast and prostate cancers[58–60] and, accordingly, we have shown that rescuing GATA3 wild-type functions in mutant cells causes a reduction in S6 phosphorylation and abolishes the effects of MDM2 inhibition on the mTOR signaling cascade. It has also been reported that upon adaptation to hormone deprivation, breast cancer cells rely heavily on PI3K signaling and that inhibition of PI3K and mTOR induces apoptosis in these cells[61]. Furthermore, our model is also supported by the observed synergistic effect of dual MDM2 and PI3K/Akt/mTOR inhibition[62,63]. Our hypothesis, however, may only partially explain the synthetic lethality between GATA3 and MDM2. Further studies are required to fully dissect the mechanism of action.

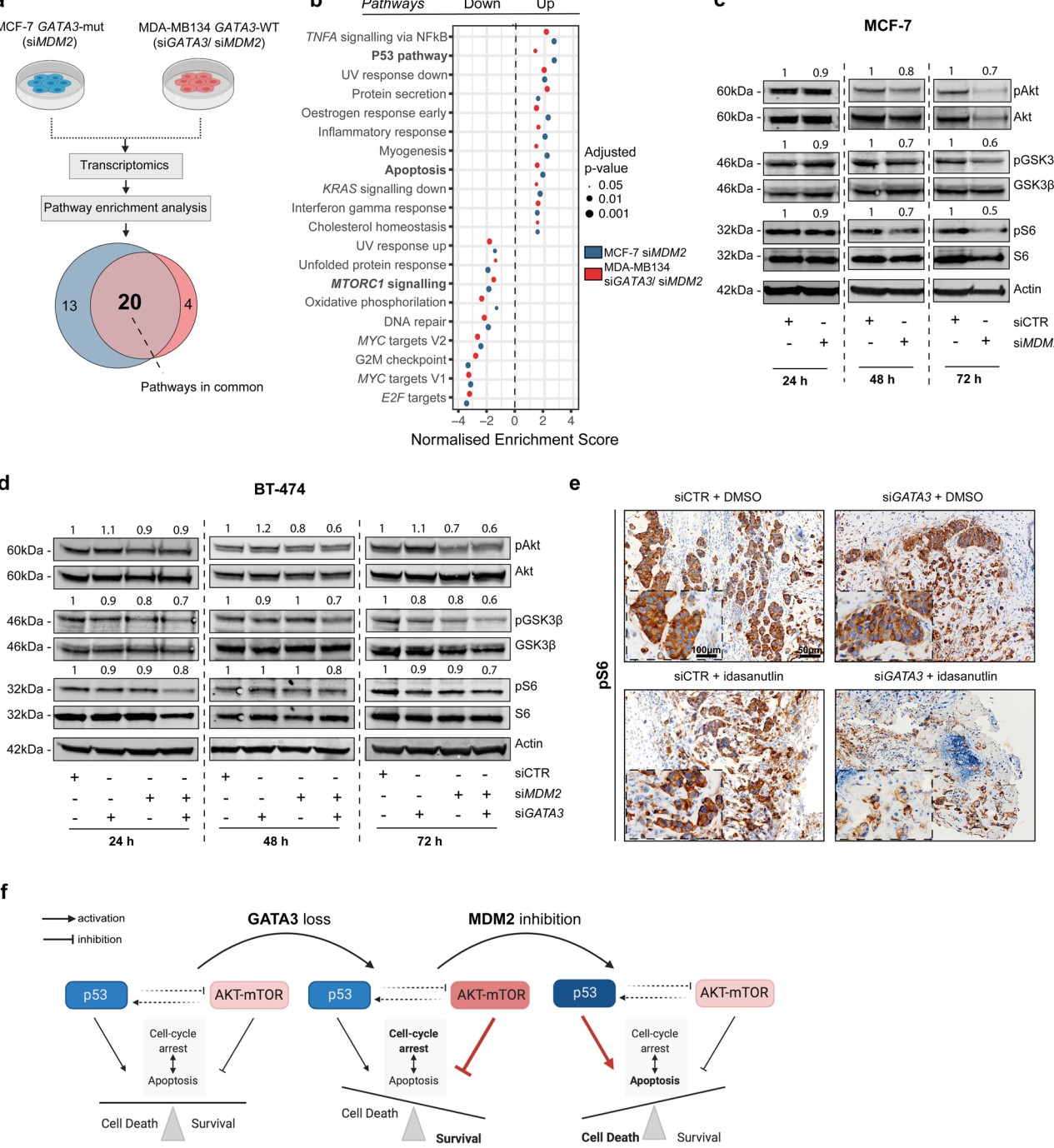

**Fig. 6 The synthetic lethality between *GATA3* and *MDM2* acts via the PI3K-Akt-mTOR signaling pathway. a** Schematic representation of the RNA-seq experimental setup to identify gene expression changes induced by concurrent *GATA3* loss and *MDM2* inhibition. Venn diagram shows the number of pathways enriched in both MCF-7 with *MDM2* siRNA and MDA-MB134 with *GATA3* siRNA and *MDM2* siRNA. **b** Normalized enrichment scores of significantly up- and down-regulated pathways identified by gene set enrichment analysis in both MCF-7 and MDA-MB134. The size of the dots is proportional to the adjusted p-value as indicated in the legend. **c, d** Immunoblot showing markers of mTOR signaling pathway activation at 24, 48, and 72 h post-siRNA transfection in **c** MCF-7 cells upon *MDM2* silencing and **d** BT-474 cells upon *GATA3* and/or *MDM2* silencing (see also Supplementary Fig. 6a, c, d). For all the western blots, quantification is relative to the loading control (actin) and normalized to the corresponding siCTR. **e** Representative immunohistochemistry micrographs of phospho-S6 stainings in BT-474 tumors extracted four days post-implantation in the CAM model (see also Supplementary Fig. 6e). **f** Schematic representation of the mechanistic hypothesis. Scale bars: **e** 50 and 100 µm. Statistical significance was determined for **b** by *fgsea*. **a, f** were created with BioRender.com.

Our findings have important clinical implications for several subsets of ER-positive breast cancer. As we have shown MDM2 inhibitors to be effective in inhibiting cell proliferation of *ESR1*-mutant breast cancer cells, and hampering tumor growth of ER-positive breast cancer PDXs resistant to fulvestrant, we believe that MDM2 inhibition might represent an effective alternative therapeutic option for breast cancers refractory to anti-estrogen therapy, often observed among *GATA3*-mutant breast cancers. Furthermore, our findings suggest that *GATA3* mutations may drive resistance to hormonal therapy by upregulating the PI3K/Akt/mTOR pathway, suggesting that inhibition of MDM2 may help overcome the resistance to PI3K/Akt/mTOR inhibition and/or to endocrine therapy in *GATA3*-deficient breast cancer. The mutual exclusivity between genetic alterations in *GATA3* and genes in the PI3K pathway suggests the PI3K inhibitors may be effective in the context of *GATA3* mutations. It would be clinically relevant to test if the synergistic effect of dual MDM2 and PI3K/Akt/mTOR inhibition is even stronger in the context of *GATA3* mutation. Third, given that aberrant activation of the PI3K pathway has been implicated in resistance to HER2-targeted therapy[64], one might hypothesize that *GATA3* mutations may be a mechanism of resistance to HER2-targeted therapy and that MDM2 inhibitors may act synergistically with trastuzumab in ER+/HER2+, *GATA3*-mutant breast cancers.

Although *TP53* mutations are a major driver of resistance to MDM2 inhibitors[65,66], very few *GATA3*-mutant ER-positive breast cancers harbor *TP53* mutations. Thus the presence of *TP53* mutations is not expected to preclude the use of MDM2 inhibitors in the vast majority of these patients. With multiple MDM2 inhibitors in clinical trials, our findings allow the rational design of clinical trials to evaluate the in-patient efficacy of MDM2 inhibitors and to specifically evaluate GATA3 status as a predictive biomarker of response. Given that GATA3 loss of expression has also been associated with poor prognosis in other cancer types[58,67], we expect our finding to have far-reaching implications beyond ER-positive breast cancer.

Despite the profound therapeutic implications, the synthetic lethality between GATA3 and MDM2 had never been reported. This unexpected finding was the result of the availability of large-scale, unbiased screening of genetic interactions in a large panel of cell lines[15] as well as a statistical algorithm[18] powerful enough to detect such interaction even when the number of cell lines harboring *GATA3* mutation is small ($n = 2$). Our study exemplifies how perturbation screens can lead to pre-clinical hypotheses that can be rapidly tested and translated into therapeutic candidates.

## Methods

**Cell lines**. ER-positive breast cancer cell lines MCF-7 (*GATA3*-mutant p.D335Gfs; *TP53* wild-type), BT-474 (*GATA3* wild-type, *TP53*-mutant p.E285K with retained transactivation activity[68]), MDA-MB134 (*GATA3* wild-type; *TP53* wild-type) and T-47D (*GATA3* wild-type, *TP53* mutant p.L194F) were kindly provided by Dr. Rachael Natrajan from The Institute of Cancer Research (London, UK), authenticated by short tandem repeat profiling. All cell lines were monitored regularly for mycoplasma contamination by PCR using specific primers as described previously[69]. All cell lines were maintained under the condition as recommended by the provider. Briefly, all cell lines were cultured in DMEM supplemented with 5% Fetal Bovine Serum, non-essential amino acids, and antibiotics (Penicillin/Streptomycin). The cells were incubated at 37 °C in a humidified atmosphere containing 5% CO2. Exponentially growing cells were used for all in vitro and in vivo studies.

MCF-7 cell lines with knock-in mutations in the *ESR1* gene (p.Y537S and p.D538G) were provided by Dr. Jeselsohn[32]. Both p.D538G and p.Y537S knock-in MCF-7 cell lines have been extensively characterized. *ESR1*-mutant cells were shown to exhibit ligand-independent growth when treated with estradiol (E2, Fig. 2 of ref. [34]) and to display resistance against selective estrogen receptor modulators (SERMs) and selective estrogen receptor degraders (SERDs, Fig. 3 of ref. [34]). Both p.Y537S and p.D538G knocked-in MCF-7 cells were shown to display a significant growth advantage in hormone-depleted conditions compared to the *ESR1*-WT MCF-7 (Fig. 2b of ref. [33]), and p.Y537S mutant cells were shown to be resistant to both tamoxifen and fulvestrant (Fig. 2d of ref. [33]).

**Transient gene knockdown and overexpression**. Transient gene knockdown was conducted using ON-TARGET plus siRNA transfection. ON-TARGET plus SMARTpool siRNAs against human *GATA3*, *MDM2*, *TP53*, ON-TARGET plus SMARTpool non-targeting control, and DharmaFECT transfection reagent were all purchased from GE Dharmacon (Supplementary Data 3). Transfection was performed according to the manufacturer's protocol. Briefly, log-phase ER-positive breast cancer cells were seeded at approximately 60% confluence. Because residual serum affects the knockdown efficiency of ON-TARGET plus siRNAs, the growth medium was removed as much as possible and replaced by serum-free medium (Opti-MEM). siRNAs were added to a final concentration of 25 nM, unless otherwise specified (Note: siRNAs targeting different genes can be multiplexed). Cells were incubated at 37 °C in 5% CO$_2$ for 24, 48, and 72 h (for mRNA analysis) or for 48 and 72 h (for protein analysis). To avoid cytotoxicity, the transfection medium was replaced with a complete medium after 24 h.

For gene overexpression, log-phase MCF-7 and BT-474 breast cancer cells were seeded in 6-well plates at approximately 60–80% confluence and transfected with control (pLV[Exp]-EGFP:T2A:Puro-CMV>Luc2 (VB190320-1059xxv)), GATA3 wild-type (pRP[Exp]-EGFP/Puro-CMV>hGATA3[NM_001002295.2] (VB201028-1114vqf)) or GATA3 mutant (p.D335Gfs) (pRP[Exp]-EGFP/Puro-CMV>{hGATA3*(c.1006dup)} (VB210314-1087tvn)) expression vectors using JetPrime buffer and reagent (PolyPlus #101000027 and #201000003, Supplementary Data 3). Eight hours after transfection, the antibiotic-free medium was replaced with a complete medium.

**RNA extraction and relative expression by qRT-PCR**. Total RNA was extracted from cells at 75% confluence using TRIZOL (Supplementary Data 3) according to the manufacturer's guidelines. cDNA was synthesized from 1 µg of total RNA using SuperScript™ VILO™ cDNA Synthesis Kit. All reverse transcriptase reactions, including no-template controls, were run on an Applied Biosystem 7900HT thermocycler. The expression for all the genes was assessed using SYBR and all qPCR experiments were conducted at 50 °C for 2 min, 95 °C for 10 min, and then 40 cycles of 95 °C for 15 s and 60 °C for 1 min on a QuantStudio 3 Real-Time PCR System (Applied Biosystems). The specificity of the reactions was verified by melting curve analysis. Measurements were normalized using *GAPDH* level as reference. The fold change in gene expression was calculated using the standard ΔΔCt method[70]. All samples were analyzed in triplicate. A list of primers is available in Supplementary Data 3.

**Immunoblot**. Total proteins were extracted by directly lysing the cells in Co-IP lysis buffer (100 mmol/L NaCl, 50 mmol/L Tris pH 7.5, 1 mmol/L EDTA, 0.1% Triton X-100) supplemented with 1× protease inhibitors and 1× phosphatase inhibitors. Cell lysates were then treated with 1× reducing agent, 1× loading buffer, boiled, and loaded onto neutral pH, pre-cast, discontinuous SDS-PAGE mini-gel system. After electrophoresis, proteins were transferred to nitrocellulose membranes using the Trans-Blot Turbo Transfer System (Bio-Rad). The transblotted membranes were blocked for 1 h in TBST 5% milk and then probed with appropriate primary antibodies (from 1:200 to 1:1000) overnight at 4 °C. A list of antibodies and working concentrations are available in Supplementary Data 3. Next, the membranes were incubated for 1 h at room temperature with fluorescent secondary goat anti-mouse (IRDye 680) or anti-rabbit (IRDye 800) antibodies (both from LI-COR Biosciences). Blots were scanned using the Odyssey Infrared Imaging System (LI-COR Biosciences) and band intensity was quantified using ImageJ software. The ratio of proteins of interest/loading control in idasanutlin-treated samples was normalized to their DMSO-treated control counterparts. All experiments were performed and analyzed in triplicate.

**Drug treatment**. In all, $10 \times 10^3$ exponentially growing cells were plated in a 96-well plate. After 24 h, cells were treated with serial dilution of RG7388-idasanutlin, RAIN-32, MI-733 (Supplementary Data 3), or dimethyl sulfoxide (DMSO). DMSO served as the drug vehicle, and its final concentration was no more than 0.1%. Cell viability was measured after 72 h using CellTiter-Glo Luminescent Cell Viability Assay reagent. Results were normalized to the vehicle (DMSO).

For the treatment experiments of the PDOs, PDOs were plated as single cells in a 96-well plate at a density of $1 \times 10^4$ cells in 10 µl Matrigel droplets. Prior to treatment, cells were allowed to recover and form organoids for 2 days. At day 3, idasanutlin, RAIN-32, or MI-733 at different dilutions was added to the medium, and cell viability was assessed after 5 days using CellTiter-Glo 3D reagent (Supplementary Data 3). Luminescence was measured on Varioskan Microplate Reader (ThermoFisher Scientific). Results were normalized to DMSO control. All experiments were performed in triplicate. Results are shown as mean ± SD. Curve fitting was performed using Prism (GraphPad) software and the nonlinear regression equation.

**Proliferation assay**. For cell lines, cell proliferation was assayed using the xCELLigence system (RTCA, ACEA Biosciences) as previously described[71]. Background impedance of the xCELLigence system was measured for 12 s using 50 µl of room temperature cell culture media in each well of E-plate 16. Cells were grown and expanded in tissue culture flasks as previously described[71]. After reaching 75% confluence, cells were washed with PBS and detached from the flasks

using a short treatment with trypsin/EDTA. 5000 cells were dispensed into each well of an E-plate 16. Cell growth and proliferation were monitored every 15 min up to 120 h via the incorporated sensor electrode arrays of the xCELLigence system, using the RTCA-integrated software according to the manufacturer's parameters. In the case of transient siRNA transfection, cells were detached and plated on xCELLigence 24 h post-transfection. For all the experiments with idasanutlin (RG7388), the drug or DMSO was added to the cells 24 h post-seeding on the xCELLigence system, as indicated in the figures. All experiments were performed in triplicate. Results are shown as mean ± SD.

For the PDOs, cell proliferation was assayed using the Incucyte S3 Live-Cell analysis system (Sartorius). Briefly, PDOs were plated as single cells in a 96-well plate at a density of $1 \times 10^4$ cells in 10 μl Matrigel droplets and allowed to recover and form organoids for 2 days. At day 3 after seeding, the 96-well plate was placed in the Incucyte incubator where a camera automatically acquired images of each well every 8 h (up to 120 h). Kinetic curves were obtained using the Incucyte analysis software and the Spheroid analysis module. Cell proliferation was calculated as the brightfield object average area. All experiments were performed in duplicate. Results are shown as mean ± SD.

**Apoptosis analysis by flow cytometry.** Cells were collected 72 h post-siRNA transfection and 48 h post-treatment with idasanutlin (RG7388) respectively, stained with annexin V (AnnV) and propidium iodide (PI), and analyzed by flow cytometry using the BD FACSCanto II cytometer (BD Biosciences, USA). Briefly, cells were harvested after incubation period and washed twice by centrifugation (1200 g, 5 min) in cold phosphate-buffered saline (DPBS; Gibco, CO; #14040133). After washing, cells were resuspended in 0.1 ml AnnV binding buffer 1X containing fluorochrome-conjugated AnnV and PI (PI to a final concentration of 1 μg/ml) and incubated in darkness at room temperature for 15 min. As soon as possible cells were analyzed by flow cytometry, measuring the fluorescence emission at 530 nm and >575 nm. Cell states were defined as previously described[72] as follow: live cells, defined as PI (−) and Annexin V (−) cells; late apoptotic cells, defined as PI (+) and Annexin V (+) cells; early apoptotic (pre-apoptotic cells), defined as PI (−) and Annexin V (+) cells; necrotic cells, defined as PI (+) and Annexin V (−) cells. Both late and early apoptotic cells were counted as "apoptotic." Unstained cells and cells stained with PI or Annexin V alone were used in each individual experiment to compensate for the fluorescence emission of each fluorochrome in the other channel and define the gating strategy more precisely. Data were analyzed by FlowJo software version 10.5.3.

**Zebrafish xenografts.** Animal experiments and zebrafish husbandry were approved by the "Kantonales Veterinaeramt Basel-Stadt" (haltenewilligung: 1024H) in Switzerland and the experiments were carried out in compliance with ethics regulations. Zebrafish were bred and maintained as described previously[73]. The staging was done by hours post-fertilization (hpf) as described previously[74] and according to FELASA and Swiss federal law guidelines. Zebrafish wild-type Tuebingen strains were used in this study. 48 h post-siRNA transfection, GATA3-silenced and control BT-474 cells were treated for 24 h with idasanutlin (25 μM). After harvesting, cells were labeled with a lipophilic red fluorescent dye (Cell-Tracker™ CM-DiI), according to the manufacturer's instructions. Zebrafish were maintained, collected, grown and staged in E3 medium at 28.5 °C according to standard protocols[75]. For xenotransplantation experiments, zebrafish embryos were anesthetized in 0.4% tricaine at 48 h (hpf), and 200 GATA3-silenced or control BT-474 cells were micro-injected into the vessel-free area of the yolk sac. Only cells with at least 80% viability in both conditions were used for grafting. After injection, embryos were incubated for 1 h at 28.5–29 °C for recovery and cell transfer verified by fluorescence microscopy. Embryos were examined for the presence of a fluorescent cell mass localized at the injection site in the yolk sac or hindbrain ventricle. Fish harboring red cells were incubated at 35 °C as described previously[38,76]. On assay day 4, embryos were screened by fluorescence microscopy for (a) normal morphology, (b) a visible cell mass in the yolk or hindbrain ventricle, using a Zeiss SteREO Discovery V20 microscope and the number of tumor-bearing fish quantified. The screening was performed independently by two scientists. For each condition, 70–100 fish were analyzed over two experiments. Representative pictures were taken using a Nikon CSU-W1 spinning disk microscope. To assess cell proliferation, fish were furthermore dissociated into single cells as described previously[77,78] and the number of fluorescence-labeled cells was then determined using flow cytometry on a BD FACSCanto II cytometer for CM-DiI–positive cells. To obtain a number of cells sufficient to be analyzed, the lysate of 20–30 fish for each condition was pooled for analysis. Only tumor-bearing fish were pooled, so the total fluorescence could be used as a surrogate measure of the total number of tumor cells. Each experiment was repeated twice.

**Chorioallantoic membrane (CAM).** Fertilized chicken eggs were obtained from Gepro Geflügelzucht AG at day 1 of gestation and were maintained at 37 °C in a humidified (60%) incubator for nine days[79] At this time, an artificial air sac was formed using the following procedure: a small hole was drilled through the eggshell into the air sac and a second hole near the allantoic vein that penetrates the eggshell membrane. A mild vacuum was applied to the hole over the air sac in order to drop the CAM. Subsequently, a square 1 cm window encompassing the hole near the

allantoic vein was cut to expose the underlying CAM[79]. After the artificial air sac was formed, BT-474 cells growing in tissue culture were inoculated on CAMs at $2 \times 10^6$ cells per CAM, on three to four CAMs each. Specifically, 48 h post-siRNA transfection, GATA3-silenced and control BT-474 cells were treated with idasanutlin (25 μM). 24 h post-treatment, cells were detached from the culture dish with Trypsin, counted, suspended in 20 μl of medium (DMEM) and mixed with an equal volume of Matrigel. To prevent leaking and spreading of cells, a 8 mm (inner diameter) sterile teflon ring (removed from 1.8 ml freezing vials, Nunc, Denmark) was placed on the CAMs and the final mixture was grafted onto the chorioallantoic membranes inoculating the cells with a pipette inside the ring[80]. Embryos were maintained at 37 °C for 4 days after which tumors at the site of inoculation were excised using surgical forceps. Images of each tumor were acquired with a Canon EOS 1100D digital camera. Surface measurements were performed by averaging the volume (height*width*width) of each tumor using ImageJ, as previously described[81]. Total $n \geq 10$ tumors for each condition were analyzed over three independent experiments.

**Patient-derived organoid (PDO) generation and sequencing.** PDOs were derived from ER-positive breast cancer patient-derived xenografts (PDX) generated at Institut Curie (Paris, France). The two metastasis-derived PDX (patients 1 and 3 in Fig. 5A) correspond to PDX HBCx-139 (GATA3-wild-type) and HBCx-137 (GATA3-mutant) were established from spinal metastases of patients progressing on endocrine treatments, as previously described[48]. The second GATA3-mutant PDX (HBCx-169) was established from the surgical specimen obtained at mastectomy from a de novo stage IV breast cancer patient. All PDXs were luminal B subtypes and were estrogen responsive[48]. MDM2 gene expression was assessed by the Affymetrix gene expression array[48].

Briefly, upon removal from the donor, mouse tissue was placed in MACS Tissue Storage Solution (Supplementary Data 3) and immediately shipped overnight on ice. Upon arrival, the tissue was immediately processed to generate PDO as previously described[82]. Briefly, the tissue was cut into small pieces and digested in 5 mL advanced DMEM/F-12, containing collagenase IV, DNase IV, hyaluronidase V, BSA, and LY27632 (Supplementary Data 3) for 1 h and 30 min at 37 °C under slow rotation and vigorous pipetting every 15 min. The tissue lysate was then filtered through a 100 μM cell strainer, centrifuged at $300 \times g$ for 10 min, and then treated with Accutase (Supplementary Data 3) for 10 min at room temperature to dissociate the remaining fragments. After 5 min of centrifugation at $300 \times g$, the cell pellet was finally suspended with growth factor reduced Matrigel (Supplementary Data 3) and seeded as drops in a tissue-culture dish. After polymerization of Matrigel, medium supplemented with growth factors[82] was added to the cells. The medium was changed every 3 days and organoids were passaged after dissociation with 0.25% Trypsin-EDTA (Supplementary Data 3).

DNA was extracted from snap-frozen organoids pellets using the DNeasy Blood & Tissue Kit (Qiagen, Cat No./ID: 69504). Sanger sequencing was performed as previously described[83] using custom-designed primers (Supplementary Data 3).

**Mouse husbandry.** All mouse experiments were approved by and performed in accordance with the guidelines and regulations of the Animal Ethics Committee of the Association for Assessment and Accreditation of Laboratory Animal Care International. NOD/SCID female mice of 6–8 weeks of age were housed under pathogen-free conditions in individually ventilated cage (IVC) systems at constant temperature and humidity at the animal facilities of Crown Bioscience, Inc.

**Primary GATA3p.D335Gfs breast cancer xenotransplantation and RAIN-32 treatment.** Clinical characteristics and immunophenotypic features of the BR5496 tumor sample used to develop these xenograft models are reported in Supplementary Table 1. PDX tumor fragments of 2–3 mm in diameter were sub-cutaneously xenografted into NOD/SCID mice. Each mouse was inoculated subcutaneously at the right flank region with BR5496 tumor chunk for tumor development. 40 mice were enrolled in the study. The PDX-inoculated mice were selected and randomly categorized into vehicle, fulvestrant, RAIN-32 50 mg/kg or RAIN-32 100 mg/kg groups (eight mice per group) when the mean tumor size reached approximately 144 mm3. The treatments started when the mean tumor size reached approximately 150 mm³ and lasted for 29 days. Randomization was performed based on the "Matched distribution" method using the StudyDirector™ software, version 3.1.399.19 randomized block design. Tumor volumes were measured two times per week in two dimensions using a caliper, and the volume was expressed in mm³ using the formula: $V = 0.5 \, a \times b^2$ where $a$ and $b$ are the long and short diameters of the tumor, respectively. The entire procedures of dosing as well as tumor and body weight measurement were conducted in a Laminar Flow Cabinet.

**Immunohistochemistry.** Tumors were fixed in 10% Paraformaldehyde (PFA) immediately after excision from the CAM. PFA-fixed and paraffin-embedded tumors were cut into 3.5 μm-thick sections. Hematoxylin and eosin (H&E) staining was performed according to standard protocols. Tissue sections were rehydrated and immunohistochemical staining was performed on a BOND-MAX immuno-histochemistry robot (Leica Biosystems) with BOND polymer refine detection solution for DAB, using anti-GATA3, cleaved caspase 3, phospho-Akt or phospho-

S6 (Supplementary Data 3) primary antibodies as substrate. Estrogen receptor immunostain was performed as described previously[84]. Photomicrographs of the tumors were acquired using an Olympus BX46 microscope. All stained sections were evaluated blindly by two independent pathologists.

**RNA sequencing and pathway analysis.** Biological duplicates were generated for all the samples analyzed. Total RNA was extracted from cells at 75% confluence using TRIZOL (Supplementary Data 3) according to the manufacturer's guidelines. RNA samples were treated with Turbo DNase (AM 1907, Thermo Fisher Scientific) and quantified using a Qubit Fluorometer (Life Technologies). RNA integrity was measured using the Agilent Bioanalyzer 2100 (Agilent Technologies).

Library generation was performed using the TruSeq Stranded mRNA protocol (Illumina). Paired-end RNA sequencing was performed on the Illumina NovaSeq 6000 platform using the $2 \times 100$bp protocol according to the manufacturer's guidelines. Reads were aligned to the GRCh37 human reference genome using STAR 2.7.1[85], and transcript quantification was performed using RSEM 1.3.2[86]. Genes without at least ten assigned reads in at least two samples were discarded. Counts were normalized using the median of ratios method from the DESeq2 package[87] in R version 3.6.1 (https://www.R-project.org/). Differential expression analysis was performed using the DESeq2 Wald test. Gene set enrichment analysis was performed using the *fgsea* R package[88] and the Hallmark gene set from the Molecular Signatures Database[89], using the ranked t statistics from the DESeq2 Wald test. Pathways with false discovery rate (FDR) < 0.05 were considered to be significant. Results were visualized using ggplot2[90].

**Analysis of The Cancer Genome Atlas (TCGA) data.** ER-positive breast cancer mutation annotation file for variant calling pipeline mutect2, FPKM gene expression data, and raw read counts of the TCGA BRCA project were downloaded using *TCGAbiolinks*[91] package. Tumor samples were classified as *GATA3*-mutant ($n = 122$) and *GATA3*-wild type ($n = 596$) according to the *GATA3* mutation status. Samples with *GATA3* mRNA expression in the bottom and top quartile were classified as *GATA3*-low ($n = 200$) and *GATA3*-high ($n = 204$), respectively. *edgeR* package[92] was used for differential expression analysis and the genes with low expression (<1 log-counts per million in ≥30 samples) were filtered out. Normalization was performed using the "TMM" (weighted trimmed mean) method[93] and differential expression was assessed using the quasi-likelihood F-test. Gene set enrichment analysis of all analyzed genes ranked based on signed *P* value according to the direction of the log-fold change was performed using the *fgsea* package[88]. Hallmark gene sets from Molecular Signatures Database[89] were used to identify significantly up-/down-regulated pathways. Pathways with FDR < 0.05 were considered significantly regulated.

**Analysis of mutually exclusive genetic alterations.** ER-positive breast cancer mutational data for the *GATA3, TP53, PIK3CA*, and *PTEN* genes and copy number status for *PTEN* derived from the TCGA PanCancer Atlas[1] and the METABRIC dataset[4] were obtained using cBioportal[94]. A total of 2379 samples were used for the analysis. Mutual exclusivity of somatic mutations in *GATA3, TP53, PIK3CA*, and *PTEN* and deep deletions for *PTEN* were calculated using one-sided Fisher's Exact and *P* < 0.05 was considered statistically significant.

**Statistics and reproducibility.** Statistical analyses were conducted using Prism software v8.0 (GraphPad Software, La Jolla, CA, USA). For in vitro studies, statistical significance was determined by the two-tailed unpaired Student's *t* test. For comparison involving multiple time points, statistical significance was determined by multiple Student's *t* test corrected for multiple comparisons with the Holm–Sidak method. A *P* value < 0.05 was considered statistically significant. For all figures, ns, not significant. For in vivo studies two-sided Fisher's Exact was used to compare the number of tumor-harboring fish. For the CAM assay, a two-tailed unpaired Student's *t* test was used. The statistical parameters (i.e., exact value of *n*, *P* values) have been noted in the figures. Unless otherwise indicated, all data represent the mean ± standard deviation from at least three independent experiments.

For the PDX experiments, summary statistics, including the mean and the standard error of the mean (SEM), were provided for the tumor volume of each group at each time point. Statistical analysis of differences in tumor volume among the groups was conducted using one-way ANOVA followed by individual comparisons using Games–Howell (equal variance not assumed). All data were analyzed in SPSS (Statistical Product and Service Solutions) version 16.0. *P* values were rounded to three decimal places, with the exception that raw P-values less than 0.001 were stated as *P* < 0.001. All tests were two-sided. *P* < 0.05 was considered to be statistically significant.

**Power calculation.** For the in vivo experiments, the samples size was calculated using a G*Power calculation[95]. For zebrafish experiments, assuming a difference of 20% in tumorigenic potential and type I error of 5%, 85 samples in each group would ensure >80% power to detect statistical differences between experimental groups using Fisher's exact test. Furthermore, assuming a 95% engraftment rate, 95 experiments would ensure we had a >95% probability of having 85 successful xenotransplantations.

For the CAM assay, assuming an effect size of 1.5 and type I error of 5%, 9 samples in each group would ensure >80% power to detect statistical differences between experimental groups using unpaired *t* tests. Furthermore, assuming a 95% engraftment rate, 10 experiments would ensure we had >91% probability of having 9 successful xenotransplantations.

**Reporting summary.** Further information on research design is available in the Nature Research Reporting Summary linked to this article.

## Data availability

RNA-sequencing data are available at the NCBI Sequence Read Archive (PRJNA623723). All the uncropped and unedited blot/gel images are available in Supplementary Data 4. Source data for graphs and charts are available in Supplementary Data 4.

## Code availability

Scripts used to generate the figures for the analysis of the RNA-sequencing were based on previously published materials as reported in "Methods."

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

## Acknowledgements

We would like to thank Stefano Medagli who helped with the generation of the figures. We would like to thank Dr. Rachael Natrajan for sharing breast cancer cell lines. RNA-sequencing analysis was performed at sciCORE scientific computing center at the University of Basel. This work was supported by the Swiss Centre for Applied Human Toxicology; the Swiss Cancer Research foundation [KFS-4543-08-2018 to C.K.Y.N. and KFS-4988-02-2020-R to S.P.]; The Professor Dr Max Cloëtta Foundation (Medical research positions to S.P.), the Swiss National Science Foundation [31003A_169352 to M.K.-d.J.]; the Dutch Cancer Society [KWF 2015_7599 to M.K.-d.J.]; Novartis [17B076 to M.K.-d.J.]; and the European Research Council [Synergy Grant 609883 to N.B.]. The funders had no role in study design, data collection and analysis, decision to publish, or preparation of the manuscript.

## Author contributions

C.K.Y.N. and S.P. conceived and supervised the study, interpreted the results, wrote and edited the manuscript with G.B. and M.C.-L.; G.B., M.C.-L., C.K.Y.N., and S.P. made final edits, figures and completed the paper; G.B., M.C.-L., S.T.-M., M.M., and V.P. performed molecular experiments and G.B. analyzed the data; M.K., M.D.M., M.K.-d.J., and C.L. provided the technical expertise for the in vivo experiments performed by G.B., M.C.-L., S.T.-M., and M.K.; G.B and F.P. performed the ex vivo experiments; S.S., H.M., and N.B. developed the statistical model for the analysis of the DRIVE data; C.E. and L.M.T. analyzed the histological stains and provided pathology expertise; J.G. and V.K. performed bioinformatic analysis of the RNA-sequencing and the publicly available data; R.M.J. provided the *ESR1* mutant models and provided critical input in the discussion of the results; F.-C.B. provided critical input in the discussion of the results; A.D., E. Montaudon and E. Marangoni provided the human *GATA3* wild-type and mutant tumor models and critical input in the discussion of the results; V.G.T. and R.C.D. performed the PDX in vivo experiments. All authors agreed to the final version of the manuscript.

## Competing interests

The authors declare the following competing interests: part of this study has been submitted for a patent application (applicants: the University of Basel and ETH Zürich; the name of the inventors: G.B., S.S., H.M., N.B., C.K.Y.N., and S.P. The patent application has been submitted to the European patent office; application number: EP19216550.4). V.T. and R.C.D. are employed at Rain Therapeutics Inc. G.B. is employed at Novartis NIBR; however, she worked on the manuscript when still affiliated with the University of Basel. R.J. received research funding from Pfizer and Lilly and is a consultant for Luminex. The other authors declare no competing interests.
