## [Peer Review File · Communications Biology]

Reviewers' comments:

Reviewer #1 (Remarks to the Author):

This study explores the synthetic lethality between GATA3 and MDM2. GATA3 is frequently mutated in breast cancer, and multiple studies suggest that these are driver mutations contributing to tumorigenesis and/or tumor progression. However, it remains elusive how GATA3 mutations impact breast tumor properties and whether GATA3 or its mutations could be a promising target for breast cancer therapy. The authors identified that inhibition of MDM2 in ER-positive breast cancer cells exhibits synthetic lethality with GATA3 deficiency. The topic is timely, and the observations are very intriguing, while some claims need to be experimentally validated to further support their conclusions.

Major concerns

(1) Figure 1a "the viability scores for each gene knock-down between the GATA3-mutant and GATA3-wild type cell lines"

The authors indicate that they analyzed 22 breast cancer cell lines data and compared GATA3 mutant data vs GATA3 wild-type data. However, as far as I know MCF7 cell line is the only commercially available GATA3 mutant cell line (this is also noted in the method section). Figure 1c indicates two data point. Another GATA3 mutant cell line could be KPL-1 (MCF-7 derivative). Please indicate which GATA3 mutant cell lines were used in the analysis.

(2) Figure 1b

This is the core data of this paper, but I couldn't find the gene list of this figure. Please consider reporting this information as excel or any other file format. Based on this figure, not only MDM2 but also other genes (~8 genes) show significant viability scores. So, it is not clear why the authors only chose MDM2 for the rest of the experiments.

(3) MCF-7 harbors the GATA3 frameshift mutation p.D335Gfs 19, a loss-of function mutation that has been recurrently observed in breast cancer patients

Multiple papers (Hruschka N, et al. Oncogene 2020; Emmanuel N, et al. Anticancer Res 2018; Takaku M, et al, Nat Commun 2018; Gustin J, et al. Oncotarget 2017; Mair B, et al. 2016 PLoS Genet; Adomas A, et al. BMC Cancer 2014) indicate these frame-shift mutations have active functions. Please clarify this statement.

(4) Figure 1

The authors concluded that MDM2 inhibition in GATA3 mutant background provides synthetic lethality. However, it is not clear if the differential sensitivity is derived from different genetic background of breast cancer cell lines. Does the overexpression (or CRISPR knockin) of D335fs GATA3 mutant in BT-474 or MDA-MB-134 cells induce hypersensitivity against MDM2 knockdown or inhibitor?

(5) FACS analysis

The authors provided only one image of the FACS data. Based on this FACS data, it seems to be very challenging to clearly separate apoptotic cells from live cells. Please explain how the authors defined these cell states.

(6) Figure 2g

As compared to MCF-7 (Fig 2c), these blots don't seem to support the conclusion of increased apoptosis level in [siGATA3 + idasanutlin] cells. Please provide more quantitative data with statistical values.

(7) Figures 3b-d

I wasn't sure how this xenograft experiment is relevant to in vitro cell proliferation assay or in vivo tumor formation or growth. The authors grafted siRNA and/or MDM2 inhibitor pretreated cells. Presumably, many of these cells are already in apoptotic state. So, lower % of tumor formation could be simply due to higher % of apoptotic cells when they were injected.

Tumor formation analysis and cell count were performed 4 days after injection. How long do siRNAs and idasanutlin express their activities?

Fig 3d shows % of fluorescent positive cells. But to support figure 3c data, it would be important to report the comparison of total tumor cell numbers in tumor positive zebrafishes, because tumor formation efficacy is different in each condition.

Please provide the p-value of [siCTR + DMSO vs [siGATA3 + idasanutlin], as idasanutlin and siGATA3 treatment slightly increased tumor formation.

(7) Figures 3g

siGATA3 alone clearly increased the tumor volume, which was also observed in zebrafish experiments. Considering the potential risk of GATA3 silencing, it would be important to note this observation.

(8) Figure 5

The authors tested two different GATA3 mutant tumors (H433fs and S410fs) and observed a similar phenotype. Presumably, these are not truncation mutations. Please describe the impacts of these mutations. It is also important to confirm these mutants by western blot. All of the experiments are based on loss of GATA3 function. Therefore, if these are not truncation mutations, do the authors have any evidence that these mutations lead to GATA3 functional loss?

Please also explain how the authors choose the control bone metastatic breast cancer tumors. Is the expression level of MDM2 similar between these organoids? Are they luminal A or B subtype?

(9) Figures 6c-d

Please provide statistical significance of these differences.

Minor concerns

(1) Citations

The frequent mutations in GATA3 were first described by Dr. Perou's group (Oncogene, 2004, doi: 10.1038/sj.onc.1207966) followed by TCGA breast cancer report (Nature, 2012, doi: 10.1038/nature11412). The significance of GATA3 function in endocrine resistant breast tumors was also recently reported by Dr Liu's group (Nat Cell Biol, 2020, doi: 10.1038/s41556-020-0514-z). The authors also emphasized that the significance of GATA3 mutations in breast cancer context. But none of the papers that described the function of GATA3 mutants were cited in the manuscript (some of them are mentioned above). In particular, multiple groups have already investigated the function of the D335Gfs GATA3 mutation. Please consider citing these papers.

(2) Figure 4d: Y-axis is missing

(3) Figure 5b: there is a typo

Reviewer #2 (Remarks to the Author):

Review:

The paper by Bianco et al, provides for the first time the synthetic lethal relationship between GATA3 and MDM2 in ER-positive breast cancers. Deletion and pharmacological inhibition of MDM2 impaired tumor growth in several models where GATA3 was knocked down or mutated. The authors have done a great job to demonstrate the synthetic lethality between GATA3 and MDM2. Several points are shown below that could help solidify their model and conclusions:

1) Figure 1. It would be great to present a diagram of GATA3 mutations/alterations in ER+ breast cancer using MSK-IMPACT, TCGA, or Metabric databases (for example). What % of these mutations are indeed predicted to be loss of function?

- 2) Please provide an excel sheet of some of the most common mutations (PI3K/GATA3/FOXA1/ER/ARID1A) found in the 22 breast cancer cell lines.
- 3) Authors should consider putting back to rescue GATA3 in MCF7 cells and observe the cell growth effects of MDM2 inhibition.
- 4) Figure 2. It would be great to show that the MCF7 cell lines with KI of ESR1 mutants are indeed endocrine therapy resistant.
- 5) Very good and impressive models to study tumor growth differences in vivo are used. However, one cannot help but wonder how come tumor xenografts of cell lines they established and upon MDM2 inhibitors or better yet patient derived xenografts that harbor GATA3 mutant have not been considered to be used.
- 6) What cohorts were used to analyze the frequencies of GATA3/TP53 or GATA3/PIK3CA mutually exclusivity? Please mention these details in the results section or figure.
- 7) What protocol did the authors use to establish the PDO? I assume it was reference 68 from Hans Clevers group. However, reference 67 is mentioned? Is this a typo? Moreover, apart from ER expression via IHC, it would be great to understand that these organoids are indeed estrogen responsive. For instance, do ER target genes get downregulated upon fulvestrant, or upregulated upon estrogen?
- 8) The mechanism linking the synthetic lethality between GATA3/MDM2 via the PI3K/AKT/mTOR pathway seems a bit weak and seems more of a strong correlation.

Reviewer #3 (Remarks to the Author):

In this study, the authors identified a new synthetic lethal interaction in human breast cancer. The authors showed that in estrogen receptor (ER)-positive breast cancers, mutations of GATA3 renders tumors to be synthetic lethal to MDM2 inhibition. Using a novel algorithm recently developed by the authors and through analysis of the breast cancer cell line (n=22) data from the large-scale, deep RNAi screen project DRIVE, the authors identified MDM2 as the top gene whose knock-down significantly reduced cell viability in the two GATA3-mutant breast cancer cell lines. The authors then went on to perform extensive in vitro and in vivo investigations to demonstrate that while mutation of GATA3 genes more ER+ breast cancer cells to become more proliferative and more tumorigenic, inhibition of MDM2 effectively reduces cell viability of ER+ breast cancer cells with GATA3 mutation in vitro and tumor formation in vivo. The translational potential of the synthetic lethal interaction of GATA3 mutation and MDM2 inhibition is further demonstrated using patient-derived organoids. Overall, this study has discovered a new synthetic lethal interaction in ER+ human breast cancer and has immediate translational potential. The study in general is well done and the manuscript was clearly written. It is recommended for publication in Nature Communications with the following changes/revisions.

First, while the authors have built a very convincing case that GATA3 mutation in ER+ human breast cancer cells indeed makes the cancer cells becoming more sensitive to MDM2 inhibitor idasanutlin (about 5-times more sensitive in different assays) in vitro, MDM2 inhibitor did not stop tumor formation in zebrafish model (Figure 3a-d). Instead, it merely reduced the number of tumors. One would argue if GATA3 mutation in ER+ human breast cancer cells is truly synthetic lethal with MDM2 inhibition, MDM2 inhibitor should be able to completely stop tumor formation. Additionally, MCF-7 cells harbor GATA3 mutation but are not very sensitive to MDM2 inhibition.

While most of the experiments were well designed, the use of very high concentrations of idasanutlin (~25 μ M) in some of the assays is problematic for data interpretation. At concentrations over 10 μ M, idasanutlin probably would have some off-target effects, in addition to MDM2 inhibition. Idasanutlin has a binding affinity in low nanomolar concentrations, there is no reason to use such high concentrations for the purpose of MDM2 inhibition. In addition, there are other clinical-grade MDM2 inhibitors (e.g. SAR405838 (MI-773), which are available commercially and could be used to further confirm that the effects observed for Idasanutlin were through MDM2 inhibition, instead of off-target activities.

We are delighted that the Reviewers and the Editorial Board of Communication Biology found our manuscript meritorious. We would like to thank all the reviewers for their constructive comments and suggestions that have helped improve our manuscript and given us a valuable opportunity to clarify our interpretation of the results. In brief, in the revised version of the manuscript, we have validated our results using additional small molecule and clinically-available MDM2 inhibitors (MI-733 and RAIN-32/milademetan), and we have shown the efficacy of MDM2 inhibition in stopping tumor growth in the PDX model of an ER-positive GATA3 mutant breast cancer resistant to fulvestrant. Additionally, with two different rescuing experiments (re-expression of wild-type GATA3 in mutant cells, and expression of mutant GATA3 in wild-type cells), we have proved that is the GATA3 mutational status, rather the different genetic backgrounds of breast cancer cells, to drive differential sensitivity to MDM2 inhibition. Lastly, presence of a mutant GATA3 is a prerequisite of mTOR signalling inhibition driven by MDM2 inhibitors, and MDM2 inhibition is effective in the context of both truncating and elongating GATA3 mutations.

Below you will find a point-by-point response to the reviewers' comments:

Reviewers' comments:

Reviewer #1 (Remarks to the Author):

This study explores the synthetic lethality between GATA3 and MDM2. GATA3 is frequently mutated in breast cancer, and multiple studies suggest that these are driver mutations contributing to tumorigenesis and/or tumor progression. However, it remains elusive how GATA3 mutations impact breast tumor properties and whether GATA3 or its mutations could be a promising target for breast cancer therapy. The authors identified that inhibition of MDM2 in ER-positive breast cancer cells exhibits synthetic lethality with GATA3 deficiency. The topic is timely, and the observations are very intriguing, while some claims need to be experimentally validated to further support their conclusions.

Authors: We would like to thank the reviewer for the positive assessment of the study.

Major concerns

(1) Figure 1a “the viability scores for each gene knock-down between the GATA3-mutant and GATA3-wild type cell lines”

The authors indicate that they analyzed 22 breast cancer cell lines data and compared GATA3 mutant data vs GATA3 wild-type data. However, as far as I know MCF7 cell line is the only commercially available GATA3 mutant cell line (this is also noted in the method section). Figure 1c indicates two data point. Another GATA3 mutant cell line could be KPL-1 (MCF-7 derivative). Please indicate which GATA3 mutant cell lines were used in the analysis.

Authors: We thank the reviewer for this point. Indeed the two GATA3-mutant cell lines were MCF-7 and KPL-1. We have now indicated this in the **Results** and provided the complete list of 22 cell lines included in the DRIVE data in **Supplementary Table 1**.

Page 5 paragraph 1 (Results)

To identify synthetically lethal vulnerabilities of *GATA3* in breast cancer, we analyzed the breast cancer cell line (n=22, **Supplementary Table 1**) data from the large-scale, deep RNAi screen project DRIVE¹ using our recently developed SLIdR algorithm². SLIdR uses rank-based statistical tests to

compare the viability scores for each gene knock-down between the *GATA3*-mutant and *GATA3*-wild type cell lines (**Fig. 1b**) and identified 13 synthetic lethal partners of *GATA3* (FDR<0.05, **Supplementary Table 2**). We interrogated the candidates for well-developed drug targets and identified *MDM2* as the top druggable gene whose knock-down significantly reduced cell viability in the two *GATA3*-mutant breast cancer cell lines (MCF-7 and KPL-1, **Fig. 1c-d**).

Page 34 (Supplementary Table and Figure legends)

Supplementary Table 1: List of 22 breast cancer cell lines included in the DRIVE analysis and their mutations in *GATA3*, *PIK3CA*, *PIK3R1*, *PTEN*, *FOXA1*, *ESR1* and *ARID1A*.

(2) Figure 1b

This is the core data of this paper, but I couldn't find the gene list of this figure. Please consider reporting this information as excel or any other file format. Based on this figure, not only *MDM2* but also other genes (~8 genes) show significant viability scores. So, it is not clear why the authors only chose *MDM2* for the rest of the experiments.

Authors: We thank the reviewer for this suggestion. In the revised version of the manuscript we have now added **Supplementary Table 2** containing the results from the analysis of the synthetic lethal partners of *GATA3*. We identified 13 significant genes (FDR<0.05). We were interested in a *GATA3* synthetic lethal partner for which effective drugs are in use, preferably in the clinical setting. Of the significant genes, *MDM2*, *ESR1*, *MDM4*, *USP7*, *FNTA* and *WIF1* are druggable according to DGIdb (Drug-gene interaction database). Hormonal therapy is already standard of care for ER+ breast cancer patients hence we excluded *ESR1* (encoding ER). Of the remaining candidates, *MDM2* is the most well developed therapeutic target. Hence we chose *MDM2* as the focus of the study. This point have been clarified in the revised version of the manuscript as follows:

Page 5 paragraph 1 (Results)

SLIdR uses rank-based statistical tests to compare the viability scores for each gene knock-down between the *GATA3*-mutant and *GATA3*-wild type cell lines (**Fig. 1b**) and identified 13 synthetic lethal partners of *GATA3* (FDR<0.05, **Supplementary Table 2**). We interrogated the candidates for well-developed drug targets and identified *MDM2* as the top druggable gene whose knock-down significantly reduced cell viability in the two *GATA3*-mutant breast cancer cell lines (MCF-7 and KPL-1, **Fig. 1c-d**).

Page 34 (Supplementary Table and Figure legends)

Supplementary Table 2: Table of SLIdR results and effect sizes for all possible synthetic lethal (SL) partners of *GATA3* from Project DRIVE. For each possible SL partner, SLIdR uses one-sided statistical tests based on the Irwin-Hall distribution to test whether the viabilities of *GATA3* mutated cell lines from knockdown of the SL partner gene are lower than expected by chance. "Mean viability of *GATA3*-mutant cell lines" and the "Mean viability of *GATA3*-wild-type cell lines" are the average viabilities from the knockdown of each possible SL partner in *GATA3* mutated and WT cell lines, respectively. The effect size is under the "Difference in viabilities" column. The resulting p-values are reported in column "p value" with the false discovery rate shown under "FDR".

(3) MCF-7 harbors the *GATA3* frameshift mutation p.D335Gfs 19, a loss-of function mutation that has been recurrently observed in breast cancer patients

Multiple papers (Hruschka N, et al. Oncogene 2020; Emmanuel N, et al. Anticancer Res 2018; Takaku M, et al, Nat Commun 2018; Gustin J, et al. Oncotarget 2017; Mair B, et al. 2016 PLoS Genet; Adomas A, et al. BMC Cancer 2014) indicate these frame-shift mutations have active functions. Please clarify this statement.

Authors: We thank the reviewer for the comment. Indeed the function of GATA3 mutations is not yet fully understood. While Hruschka et al. and Takaku et al studied the X308_Splice and the R330fs zinc finger mutations but not the functionally distinct p.D335Gfs, it is true that multiple papers have shown that the p.D335Gfs mutation has some dominant-negative effect rather than being a simple loss-of-function mutation^{3,4}. We referred to it as a loss-of-function mutation as this mutation has been shown to disrupt binding to GATA motifs in DNA⁴, but we agree with the reviewer that this nomenclature is outdated as the same mutant displays gain-of-function properties such as increased protein stability as well⁴. We have therefore amended the **Results** section as follows:

Page 5 paragraph 2 (Results)

MCF-7 harbors the *GATA3* frameshift mutation p.D335Gfs⁵, a truncating mutation recurrently observed in breast cancer patients^{6,7} and that has been reported to have both loss and gain-of-function effects, and specifically acts as a dominant negative mutant with lower DNA binding affinity but increased half-life⁴.

(4) Figure 1

The authors concluded that MDM2 inhibition in *GATA3* mutant background provides synthetic lethality. However, it is not clear if the differential sensitivity is derived from different genetic backgrounds of breast cancer cell lines. Does the overexpression (or CRISPR knockin) of D335fs *GATA3* mutant in BT-474 or MDA-MB-134 cells induce hypersensitivity against MDM2 knockdown or inhibitor?

Authors: We thank the reviewer for the comment. Following the reviewer's question, we overexpressed *GATA3_D335fs* in BT474 wild-type cells (**Supplementary Figure 2h**). Notably, BT474 cells overexpressing *GATA3_D335fs* significantly proliferated more compared to control cells (**Figure 2h**). Similar to the phenotype induced by *GATA3* gene silencing, overexpression of *GATA3_D335G* protein sensitized cells to treatment with RG7388-idasanutlin. Indeed, while idasanutlin did not significantly affect cell viability in control BT474 cells, it significantly reduced the viability of *GATA3_D335fs* overexpressing cells (**Figure 2h**). These data further support our hypothesis that *GATA3_D335fs* mutant background confers synthetic lethality with MDM2 inhibition in ER positive breast cancer cells.

The new data have been added to the revised manuscript as follows:

Page 7 paragraph 2 (Results)

To rule out the possibility that the differential sensitivity was derived from different genetic backgrounds of breast cancer cell lines, we asked whether overexpressing the p.D335Gfs mutation *GATA3*-wild-type cells would induce hypersensitivity against MDM2 inhibition (**Supplementary Fig. 2h**). Notably, BT474 cells overexpressing *GATA3* p.D335Gfs mutation significantly proliferated more compared to control cells (**Fig. 2h**). Similar to the phenotype induced by *GATA3* gene silencing, idasanutlin significantly reduced cell proliferation in *GATA3* D335Gfs-overexpressing cells but not in control cells (**Fig. 2h**). These data further support our hypothesis that the *GATA3* D335Gfs mutant background confers synthetic lethality with MDM2 inhibition in ER-positive breast cancer cells.

Page 23 paragraph 2 (Methods)

For gene overexpression, log-phase MCF-7 and BT-474 breast cancer cells were seeded in 6-well plates at approximately 60%-80% confluence and transfected with control (pLV[Exp]-EGFP:T2A:Puro-CMV>Luc2 (VB190320-1059xxv)), GATA3 wild-type (pRP[Exp]-EGFP/Puro-CMV>hGATA3[NM_001002295.2] (VB201028-1114vqf)) or GATA3 mutant (D335Gfs) (pRP[Exp]-EGFP/Puro-CMV>{hGATA3*(c.1006dup)} (VB210314-1087tvn)) expression vectors using JetPrime buffer and reagent (PolyPlus #101000027 and #201000003, **Supplementary Table 3**). 8 hours after transfection, the antibiotic-free medium was replaced with a complete medium.

Page 18 (Figure 2 legend)

Fig. 2: GATA3 status determines response to MDM2 inhibitors *in vitro*. (a,d,e,h,i,j) Proliferation kinetics of (a) GATA3-mutant MCF-7 under increasing dosage of idasanutlin, (d) control and GATA3-WT rescued MCF-7 upon 12.5 μ M idasanutlin treatment, (e) BT-474 upon GATA3 silencing and/or treatment with 12.5 μ M idasanutlin, (h) BT-474 upon GATA3_D335Gfs overexpression and/or treatment with 12.5 μ M idasanutlin, (i,j) GATA3-mutant MCF-7 carrying a wild-type *ESR1* or mutant *ESR1* (p.D538G/p.Y537S) upon treatment with 12.5 μ M idasanutlin. ... Data are mean \pm s.d. (a,b,d,e,g,h,i,j,k) $n \geq 3$ replicates. Statistical significance was determined for (a,b,d,e,g,h,i,j,k) by the two-tailed unpaired Student's t-test.

Page 35 (Supplementary Figure 2 legend)

Supplementary Fig. 2: GATA3 status determines response to MDM2 inhibitors *in vitro*. ... (h) Immunoblot of GATA3 in BT-474 cells 48 and 72 hours post transfection with control or GATA_D335Gfs vector.

(5) FACS analysis

The authors provided only one image of the FACS data. Based on this FACS data, it seems to be very challenging to clearly separate apoptotic cells from live cells. Please explain how the authors defined these cell states.

Authors: Based on a previously described method⁸, we gated the cells as follows: live cells, defined as PI (-) and Annexin V (-) cells; late apoptotic cells, defined as PI (+) and Annexin V (+) cells; early apoptotic (pre-apoptotic cells), defined as PI (-) and Annexin V (+) cells; necrotic cells, defined as PI (+) and Annexin V (-) cells. Both late and early apoptotic cells were counted as "apoptotic". Additionally, unstained cells and cells stained with PI or Annexin V alone were used in each individual experiment to compensate for the fluorescence emission of each fluorochrome in the other channel and define the gating strategy more precisely. We have added the information to the **Methods**.

Page 26 paragraph 1 (Methods)

Apoptosis analysis by flow cytometry

Cells were collected 72 hours post-siRNA transfection and 48 hours post-treatment with idasanutlin (RG7388) respectively, stained with annexin V (AnnV) and propidium iodide (PI), and analysed by flow cytometry using the BD FACSCanto II cytometer (BD Biosciences, USA). Briefly, cells were harvested after incubation period and washed twice by centrifugation (1,200g, 5min) in cold phosphate-buffered saline (DPBS; Gibco, CO; #14040133). After washing, cells were resuspended in 0.1ml AnnV binding buffer 1X containing fluorochrome-conjugated AnnV and PI (PI to a final concentration of 1 μ g/ml) and incubated in darkness at room temperature for 15min. As soon as possible cells were analyzed by flow cytometry, measuring the fluorescence emission at 530 nm and >575 nm. Cell states were defined as previously described⁸ as follow: live cells, defined as PI (-) and

Annexin V (-) cells; late apoptotic cells, defined as PI (+) and Annexin V (+) cells; early apoptotic (pre-apoptotic cells), defined as PI (-) and Annexin V (+) cells; necrotic cells, defined as PI (+) and Annexin V (-) cells. Both late and early apoptotic cells were counted as “apoptotic”. Unstained cells and cells stained with PI or Annexin V alone were used in each individual experiment to compensate for the fluorescence emission of each fluorochrome in the other channel and define the gating strategy more precisely. Data were analyzed by FlowJo software version 10.5.3.

(6) Figure 2g

As compared to MCF-7 (Fig 2c), these blots don't seem to support the conclusion of increased apoptosis level in [siGATA3 + idasanutlin] cells. Please provide more quantitative data with statistical values.

Authors: We understand the reviewer's concerns as the differences in the apoptotic protein levels are weaker compared to the MCF-7 cells immunoblot. We believe the weaker effect is due to the fact that *GATA3* silencing is only partial and transient, while MCF-7 cells constitutively express *GATA3* mutant and, according to our hypothesis, are therefore more sensitive to the drug. Additionally, while the strongest effects were obtained at the highest dose in MCF-7 cells, only the lower dose (12.5 μ M) was used to treat the BT-474 cells. As we agree with the reviewer that this immunoblot is not as informative as the one for MCF-7 cells, we have moved the blot to **Supplementary Figure 2c**. We believe we have provided sufficient evidence of the idasanutlin-induced apoptosis in BT-474 cells using additional two assays, complementary to immunoblot: AnnexinV staining followed by FACS analysis (**Figure 2f**), Cl. caspase 3 IHC staining on BT-474 derived tumors (**Figure 3h and Supplementary Fig. 3**).

(7) Figures 3b-d

I wasn't sure how this xenograft experiment is relevant to in vitro cell proliferation assay or in vivo tumor formation or growth. The authors grafted siRNA and/or MDM2 inhibitor pretreated cells. Presumably, many of these cells are already in apoptotic state. So, lower % of tumor formation could be simply due to higher % of apoptotic cells when they were injected.

Authors: We thank the reviewer for this comment. Given that adding idasanutlin directly to the fish water is toxic to the zebrafish, grafting pretreated cells was a way to circumvent this issue. It has previously been shown that idasanutlin-induced apoptosis is delayed relative to drug exposure and does not require continuous treatment⁹. These results demonstrate that once the apoptotic component of the p53 pathway is activated by idasanutlin, continuous exposure to idasanutlin is not required to sustain this activation, but also that apoptosis is not yet induced at time of injection. Specifically, when treating cells with a single dose of idasanutlin for 16 hours followed by washout, induction of apoptosis is much stronger at 48 hours post-washout compared with 24 hours post-washout. Indeed, in our experiments we pre-treated tumor cells with idasanutlin for 24 hours, and then injected them into the fish. As an extra precaution, when collecting and counting cells for engraftment we checked for cell viability and proceeded with grafting only in cases of viability equal or higher to 80% in both conditions. We have added the information above to the **Results** and the **Methods** sections.

Page 8 paragraph 3 (Results)

Adding idasanutlin directly to the fish water is toxic to the zebrafish. Given that idasanutlin-induced apoptosis is delayed⁹, and that intermittent dosing schedules (once or twice a week) of idasanutlin induce reduction in mean tumor volume compared with continuous dosing⁹, we circumvented fish toxicity by pre-treating *GATA3*-silenced and control BT-474 cells with idasanutlin (25 μ M) or vehicle

(DMSO) for 24 hours, and 48 hours post-siRNA transfection, followed by wash-out. Twenty-four hours post-treatment, we labeled the cells with a red fluorescent cell tracker, injected them into the yolk sac of zebrafish embryos and screened embryos for tumor cell engraftment after four days (**Fig. 3a**)¹⁰.

Page 26 paragraph 2 (Methods)

48 h post-siRNA transfection, *GATA3*-silenced and control BT-474 cells were treated for 24 h with idasanutlin (25 μ M). After harvesting, cells were labeled with a lipophilic red fluorescent dye (CellTracker™ CM-DiI), according to the manufacturer's instructions. Zebrafish were maintained, collected, grown and staged in E3 medium at 28.5°C according to standard protocols¹¹. For xenotransplantation experiments, zebrafish embryos were anesthetized in 0.4% tricaine at 48 h (hpf) and 200 *GATA3*-silenced or control BT-474 cells were micro-injected into the vessel-free area of the yolk sac. Only cells with at least 80% viability in both conditions were used for grafting. After injection, embryos were incubated for 1 hour at 28.5–29°C for recovery and cell transfer verified by fluorescence microscopy.

Tumor formation analysis and cell count were performed 4 days after injection. How long do siRNAs and idasanutlin express their activities?

Authors: We thank the reviewer for this question. Gene knockdown can be detected as early as 4 h and can last up to 5 days, and even 7 days in some cases^{12,13}. Indeed, we have shown efficient *GATA3* and *MDM2* gene knockdown up to 72h in vitro (**Supplementary Fig. 1**) and *GATA3* protein reduction on IHC performed on extracted tumors up to 7 days post siRNA transfection (**Fig. 3h**). As for the duration of idasanutlin treatment, in the PK/PD model by Higgins et al.⁹, it has been shown that intermittent dosing schedules (once or twice a week) of idasanutlin induce reduction in mean tumor volume compared with vehicle control and continuous dosing.

Fig 3d shows % of fluorescent positive cells. But to support figure 3c data, it would be important to report the comparison of total tumor cell numbers in tumor positive zebrafish, because tumor formation efficacy is different in each condition.

Authors: We agree with the reviewer that it would be informative to compare the total number of cells in tumor-positive zebrafish between the different conditions. However, the nature of the assay did not allow such an analysis. We had to combine the lysate of 20-30 fish for each condition in order to obtain sufficient cells to be analysed. However, we only pooled tumor-bearing fish, therefore one could consider the total fluorescence as a surrogate measure of the total number of tumor cells. We have further explained these points in the **Methods**.

Page 27 paragraph 2 (Methods)

To assess cell proliferation, fish were furthermore dissociated into single cells as described previously^{14,15} and the number of fluorescence-labeled cells was then determined using flow cytometry on a BD FACSCanto II cytometer for CM-DiI-positive cells. To obtain an amount of cells sufficient to be analysed, the lysate of 20-30 fish for each condition were pooled for analysis. Only tumor-bearing fish were pooled, so the total fluorescence could be used as a surrogate measure of the total number of tumor cells. Each experiment was repeated twice.

Please provide the p-value of [siCTR + DMSO vs [siGATTA3 + idasanutlin], as idasanutlin and siGATA3 treatment slightly increased tumor formation.

Authors: For **Figure 3b** the p-value of [siCTR + DMSO] vs [siGATA3 + idasanutlin] is p=0.3147 (Fisher's test). For **Figure 3d**, the p-value of [siCTR + DMSO] vs [siGATA3 + idasanutlin] is 0.7265 (Unpaired t-test). We have added these to the figure as "ns".

(7) Figures 3g

siGATA3 alone clearly increased the tumor volume, which was also observed in zebrafish experiments. Considering the potential risk of GATA3 silencing, it would be important to note this observation.

Authors: We thank the reviewer for the observation. Indeed, GATA3 loss of function has been shown to be associated with a worse prognosis in breast cancer patients, therefore our results are in line with the in-patients data. Considering the advantage in tumor growth conferred by GATA3 silencing, the effect of MDM2 inhibition is even more significant. While we understand the concerns of the reviewers in the sense of what could be the dangerous consequences of inhibiting GATA3, we would like to clarify that we are not proposing a dual GATA3 and MDM2 pharmacological inhibition, but rather the use of MDM2 inhibitors for the treatment of breast cancer patients harboring pre-existing GATA3 mutations. However, we have noted the observation in the revised manuscript as follow:

Page 9 paragraph 2 (Results)

"Notably, tumors derived from GATA3-silenced cells were significantly larger than the control counterpart (**Fig. 3g**)."

(8) Figure 5

The authors tested two different GATA3 mutant tumors (H433fs and S410fs) and observed a similar phenotype. Presumably, these are not truncation mutations. Please describe the impacts of these mutations. It is also important to confirm these mutants by western blot. All of the experiments are based on loss of GATA3 function. Therefore, if these are not truncation mutations, do the authors have any evidence that these mutations lead to GATA3 functional loss?

Authors: We thank the reviewer for the question. Indeed, those mutations are not predicted to be truncating mutations, but to generate slightly elongated proteins of 55kD for p.S410fs and 51kD for p.H433fs, respectively. As of now, no functional validation has ever been performed for these specific mutations. However, Gaynor et al. have shown that some other elongating mutations of similar larger size are less stable and less abundant than wild type GATA3 in cells¹⁶. Gaynor et al. classified GATA3 mutations into three distinct functional categories, one of which was elongating mutations. They found that the relative transcriptional activities of these GATA3 mutants were significantly reduced compared to that of the wild-type GATA3¹⁶. When co-transfected with wild type GATA3, elongated mutants have been shown to reduce the transactivation activity of the wild-type protein in a dominant negative manner¹⁶, consistent with the heterozygosity of GATA3 mutations and phenotype induced by some truncating mutations such as pD335Gfs. Following the reviewer's suggestion, we have included the GATA3 immunoblotting of all three PDX breast cancer tissues in the revised manuscript. As shown in **Supplementary Fig. 5c**, HBCx-139 (GATA3 WT) expresses one band of the expected molecular weight for GATA3 wild type protein (48kD), while HBCx-137 (metastasis, GATA3 p.S410fs) and HBCx-169 (primary, GATA3 p.H433fs) indeed show the expression of two elongated isoforms higher than 50kD. Additionally, while HBCx-137 still expresses the WT allele, no GATA3 wild type protein could be detected in HBCx-169. Although in both cases, the mutations are heterozygous, as for virtually all GATA3 mutations observed in breast cancer, our data suggest that

p.H433fs may act in a dominant negative manner thus impairing the expression and/or protein stability of the wild type protein, similar to what is already known about other mutations such as the p.D335Gfs carried by MCF7 cells. While this hypothesis requires a functional validation, and while *GATA3* truncating and elongating mutations have been previously proposed to be functionally distinct¹⁷, our data suggest that MDM2 inhibitors might be efficacious on different classes of *GATA3* somatic mutations, most probably all characterized by a loss of transactivation activity through different mechanisms.

We have added the information above to the **Results** and the **Discussion** sections, and included the results in **Supplementary Fig. 5**.

Page 10 paragraph 2 (Results)

Both p.H433fs and p.S410fs *GATA3* mutations result in elongated protein isoforms (51 and 55 kD, respectively), with the primary tumor (HBCx-169) showing no *GATA3* wild type protein expression (**Supplementary Fig. 5c**).

Page 14 paragraph 1 (Discussion)

Of note, our data suggest that MDM2 inhibitors might be efficacious on different classes of *GATA3* somatic mutations, mostly frameshift truncating and elongating mutations affecting the stability or transactivation activity of the wild-type protein through diverse mechanisms^{4,16}.

Please also explain how the authors choose the control bone metastatic breast cancer tumors. Is the expression level of MDM2 similar between these organoids? Are they luminal A or B subtype?

Authors: The PDXs have been selected based on their positivity for the estrogen receptor, and mutation profile (*GATA3* mutational status either WT or mutant, must be wild type for *TP53* and *PIK3CA*, and no *PTEN* deletion or *MDM2* amplification) among those available at the Institut Curie . All three PDX have a luminal B phenotype. These 3 models show comparable MDM2 mRNA expression levels compared to other ER+ PDXs (**Supplementary Fig. 5d**). **Among these 3 models, the *GATA3* WT HBCx-139 shows *MDM2* mRNA expression level between the two mutant models.**

The fact there is no such clear segregation between *GATA3* mutants and WT PDXs based on MDM2 expression levels indicates that responsiveness to MDM2 inhibitors is not simply the result of MDM2 amplification, but rather a specific effect of *GATA3* mutations. We have added the information above to the **Results** and the **Methods** sections, and included the results in **Supplementary Fig. 5d**.

Page 10 paragraph 2 (Results)

Both p.H433fs and p.S410fs *GATA3* mutations result in elongated protein isoforms (51 and 55 kD, respectively), with the primary tumor (HBCx-169) showing no *GATA3* wild type protein expression (**Supplementary Fig. 5c**). Gene expression analysis showed no such clear segregation between *GATA3* mutants and WT PDXs based on MDM2 expression (**Supplementary Fig. 5d**).

Page 28 paragraph 2 (Methods)

All PDXs were luminal B subtypes and were estrogen responsive¹⁸. MDM2 gene expression was assessed by the Affymetrix gene expression array¹⁸.

Page 36 (Supplementary Fig. 5 legend)

Supplementary Fig. 5: *GATA3* mutations predict response to *MDM2* inhibitors in ER-positive breast cancer PDOs and PDX. ... (c) Immunoblot of *GATA3* in tissue derived from PDXs carrying wild-type *GATA3* (HBCx-139), *GATA3* mutant p.S410fs (HBCx-137), and *GATA3* mutant p.H433fs (HBCx-169). (d) *MDM2* gene expression (log₂, Affymetrix array) in the ER-positive breast cancer PDX cohort of the Institut Curie.

(9) Figures 6c-d

Please provide statistical significance of these differences.

Authors: Following the reviewer's request, we have now included the statistics for the immunoblots in **Supplementary Figure 6b-c** (previously Figures 6c-d).

Page 37 (Supplementary Fig. 6 legend)

Supplementary Fig. 6: The synthetic lethality between *GATA3* and *MDM2* acts via the PI3K-Akt-mTOR signaling pathway. ... (b-c) Statistical analysis (two-tailed unpaired Student's t-test) of p-AKT, p-GSK3B and p-S6 relative expression from the immunoblots in Fig. 6c-d.

Minor concerns

(1) Citations

The frequent mutations in *GATA3* were first described by Dr. Perou's group (Oncogene, 2004, doi: 10.1038/sj.onc.1207966) followed by TCGA breast cancer report (Nature, 2012, doi: 10.1038/nature11412). The significance of *GATA3* function in endocrine resistant breast tumors was also recently reported by Dr Liu's group (Nat Cell Biol, 2020, doi: 10.1038/s41556-020-0514-z). The authors also emphasized that the significance of *GATA3* mutations in breast cancer context. But none of the papers that described the function of *GATA3* mutants were cited in the manuscript (some of them are mentioned above). In particular, multiple groups have already investigated the function of the D335Gfs *GATA3* mutation. Please consider citing these papers.

Authors: We thank the reviewer for the suggestion. We have now cited all the papers mentioned by the reviewer. We have also included additional papers describing the p.D335Gfs mutation such as, but not limited to, the work by Adomas et. al (BMC Cancer 2014, doi: <https://doi.org/10.1186/1471-2407-14-278>).

(2) Figure 4d: Y-axis is missing

Authors: We thank the reviewer for spotting the missing axis. We have updated the figure accordingly.

(3) Figure 5b: there is a typo

Authors: We thank the reviewer for spotting the error. We have updated the figure accordingly.

Reviewer #2 (Remarks to the Author):

Review:

The paper by Bianco et al, provides for the first time the synthetic lethal relationship between *GATA3* and *MDM2* in ER-positive breast cancers. Deletion and pharmacological inhibition of *MDM2*

impaired tumor growth in several models where GATA3 was knocked down or mutated. The authors have done a great job to demonstrate the synthetic lethality between GATA3 and MDM2. Several points are shown below that could help solidify their model and conclusions:

Authors: We would like to thank the reviewer for the positive assessment of the study.

1) Figure 1. It would be great to present a diagram of GATA3 mutations/alterations in ER+ breast cancer using MSK-IMPACT, TCGA, or Metabric databases (for example). What % of these mutations are indeed predicted to be loss of function?

Authors: We thank the reviewer for this suggestion. In the revised manuscript, we have added **Fig. 1a** summarising the *GATA3* mutations in ER+ breast cancers from the Metabric and TCGA datasets (we did not include MSK-IMPACT as ER status was not reported on cBioPortal). Of the 330 *GATA3* mutations, 294 (89%) are predicted to be ‘Drivers’ according to the OncoKb annotation (these include 178 truncating, 24 missense and 92 splice site mutations), while the remaining mutations (11%) are classified as “Variants of Unknown significant” (32 missense and 4 in-frame mutations).

The majority of *GATA3* mutations are insertion/deletion (indel) mutations that induce frame-shifts or alternative splicing, resulting in protein truncation or extension¹⁷. While such type of mutations are usually considered loss of function mutations, virtually all frameshift *GATA3* mutations observed in breast cancer are heterozygous and mutant *GATA3* transcripts and proteins are highly expressed in breast cancer. Additionally, the mutational bias toward the distal part of the protein (DNA binding and homodimer¹⁹ region) are indicative of a positive selection and strongly suggest that these mutations act in a dominant negative manner by impairing the wild-type function^{4,20}. Indeed, it has been reported that the majority of these mutants (including some missense^{21,22}) by forming homodimers with the wild-type protein cause alterations of protein stability²³, aberrant nuclear localization, decrease in transcription activation¹⁶ and loss of DNA binding²³ that result in loss of canonical *GATA3* function and reprogramming of the transcriptional network²³. All together these evidence indicate that somatic mutations in *GATA3* (frameshift and some missense) cause loss of the canonical protein function via a dominant negative mechanism.

Page 5 paragraph 1 (Results)

Most *GATA3* mutations in ER+ breast cancer introduce frameshifts or alternative splicing resulting in protein truncation or extension¹⁷, with 89% of them predicted to be driver mutations (**Fig. 1a**).

Page 13 paragraph 3 (Discussion)

GATA3 is mutated in 12-18% of breast cancer^{7,24} with predominantly frameshift mutation resulting in protein truncation or extension¹⁷. These mutations mostly act in a dominant negative manner by impairing the wild-type function^{4,20} through diverse mechanism such as alteration of protein stability²³, aberrant nuclear localization, decrease in transcription activation^{4,16} and loss of DNA binding²³, all resulting in the loss of canonical *GATA3* function and reprogramming of the transcriptional network²³.

Page 17 (Figure 1 legend)

Fig. 1: *GATA3* and *MDM2* are synthetic lethal in ER-positive breast cancer. (a) Lollipop plot depicting *GATA3* somatic mutations and OncoKB²⁵ annotation in ER+ breast cancer derived from the TCGA PanCancer Atlas⁷ and the METABRIC datasets⁶.

2) Please provide an excel sheet of some of the most common mutations (PI3K/GATA3/FOXA1/ER/ARID1A) found in the 22 breast cancer cell lines.

Authors: In the revised version of the manuscript we have added **Supplementary Table 1** summarizing the most common mutations found in the 22 breast cancer cell lines.

Page 34 (Supplementary Table 1)

Supplementary Table 1: List of 22 breast cancer cell lines included in the DRIVE analysis and their mutations in *GATA3*, *PIK3CA*, *PIK3R1*, *PTEN*, *FOXA1*, *ESR1* and *ARID1A*.

3) Authors should consider putting back to rescue GATA3 in MCF7 cells and observe the cell growth effects of MDM2 inhibition.

Authors: We thank the reviewer for the comment. Following the reviewer's suggestion, we rescued GATA3 wild-type expression in MCF-7 cells (**Figure 2d** and **Supplementary Figure 2b**). Notably, MCF-7 cells overexpressing GATA3 WT proliferated significantly less compared to control cells (**Figure 2d**). More importantly, rescuing wild-type GATA3 desensitized cells to idasanutlin. Indeed, and contrary to control cells, idasanutlin did not significantly affect cell viability in MCF-7 cells where GATA3 wild-type was overexpressed. These data, together with the ones obtained in BT474 cells overexpressing GATA3_D335Gfs, strongly support our hypothesis that GATA3_D335fs mutant background confers synthetic lethality with MDM2 inhibition in ER positive breast cancer cells.

Page 6 paragraph 3 (Results)

To demonstrate that MCF-7 sensitivity was indeed due to the presence of mutant GATA3, we rescued GATA3 wild-type expression in MCF-7 cells (**Fig. 2d** and **Supplementary Fig. 2b**). Notably, MCF-7 cells overexpressing wild-type GATA3 proliferated significantly less compared to control cells (**Fig. 2d**). More importantly, rescuing wild-type GATA3 desensitized cells to idasanutlin (**Fig. 2d**). Indeed, and contrary to control cells, idasanutlin did not significantly affect cell viability in MCF-7 cells where GATA3 wild-type was overexpressed (**Fig. 2d**).

Page 18 (Figure 2 legend)

Fig. 2: GATA3 status determines response to MDM2 inhibitors *in vitro*. (a,d,e,h,i,j) Proliferation kinetics of (a) *GATA3*-mutant MCF-7 under increasing dosage of idasanutlin, (d) control and *GATA3*-WT rescued MCF-7 upon 12.5 μ M idasanutlin treatment, (e) BT-474 upon *GATA3* silencing and/or treatment with 12.5 μ M idasanutlin, (h) BT-474 upon *GATA3*_D335Gfs overexpression and/or treatment with 12.5 μ M idasanutlin, (i,j) *GATA3*-mutant MCF-7 carrying a wild-type *ESR1* or mutant *ESR1* (p.D538G/p.Y537S) upon treatment with 12.5 μ M idasanutlin.... Data are mean \pm s.d. (a,b,d,e,g,h,i,j,k) $n \geq 3$ replicates. Statistical significance was determined for (a,b,d,e,g,h,i,j,k) by the two-tailed unpaired Student's t-test.

Page 23 paragraph 2 (Methods)

For gene overexpression, log-phase MCF-7 and BT-474 breast cancer cells were seeded in 6-well plates at approximately 60%-80% confluence and transfected with control (pLV[Exp]-EGFP:T2A:Puro-CMV>Luc2 (VB190320-1059xxv)), *GATA3* wild-type (pRP[Exp]-EGFP/Puro-CMV>hGATA3[NM_001002295.2] (VB201028-1114vqf)) or *GATA3* mutant (D335Gfs) (pRP[Exp]-EGFP/Puro-CMV>{hGATA3*(c.1006dup)} (VB210314-1087tvn)) expression vectors using JetPrime buffer and reagent (PolyPlus #101000027 and #201000003, **Supplementary Table 3**). 8 hours after transfection, the antibiotic-free medium was replaced with a complete medium.

Page 35 (Supplementary Figure 2 legend)

Supplementary Fig. 2: GATA3 status determines response to MDM2 inhibitors *in vitro*. ... (b) Immunoblot of GATA3 in MCF-7 cells 48 and 72 hours post transfection with control or GATA_WT vector. ... For all the western blots, quantification is relative to the loading control (actin) and normalized to the corresponding DMSO control.

4) Figure 2. It would be great to show that the MCF7 cell lines with KI of ESR1 mutants are indeed endocrine therapy resistant.

Authors: We thank the reviewer for the question. Both D538G and Y537S knock-in MCF7 cell lines used in our study have been previously characterized. In the work from Bahrein et al., *ESR1* mutant cells exhibit ligand-independent growth when treated with estradiol (E2) and display resistance against selective estrogen receptor modulators (*SERMs*) (Figure 2 of ²⁶) and selective estrogen receptor degraders (*SERDs*) (Figure 3 of ²⁶). Both Y537S and D538G knocked-in MCF7 cells display a significant growth advantage in hormone-depleted conditions compared to the MCF7 with WT *ESR1* (Figure 2b of ²⁷), and Y537S mutant cells were shown to be resistant to both tamoxifen and fulvestrant (Figure 2d, ²⁷). The MCF7 *ESR1* mutant cells used in our study are the same as those used in the above mentioned studies ^{26,27} and kindly provided by Dr. Jeselsohn from the Department of Medical Oncology, Dana Farber-Cancer Institute (Boston). We have updated the Results section of the revised manuscript to better clarify the knock-in cells derivation and characterization, and relative references.

Page 7 paragraph 3 (Results)

Both *ESR1*-mutant cells have previously been shown to exhibit estradiol(E2)-independent growth and resistance to fulvestrant and tamoxifen ^{26,27}.

Page 22 paragraph 2 (Methods)

MCF-7 cell lines with knock-in mutations in the *ESR1* gene (p.Y537S and p.D538G) were provided by Dr. Jeselsohn ²⁸. Both D538G and Y537S knock-in MCF-7 cell lines have been extensively characterized. *ESR1*-mutant cells were shown to exhibit ligand-independent growth when treated with estradiol (E2, Figure 2 of ²⁶) and to display resistance against selective estrogen receptor modulators (*SERMs*) and selective estrogen receptor degraders (*SERDs*, Figure 3 of ²⁶). Both Y537S and D538G knocked-in MCF-7 cells were shown to display a significant growth advantage in hormone-depleted conditions compared to the *ESR1*-WT MCF-7 (Figure 2b of ²⁷), and Y537S mutant cells were shown to be resistant to both tamoxifen and fulvestrant (Figure 2d of ²⁷).

5) Very good and impressive models to study tumor growth differences *in vivo* are used. However, one cannot help but wonder how come tumor xenografts of cell lines they established and upon MDM2 inhibitors or better yet patient derived xenografts that harbor GATA3 mutant have not been considered to be used.

Authors: We thank the reviewer for the comment and we recognize the importance of including a PDX model to further corroborate our data. Owing to the use of a transient siRNA model and regulatory hurdles to perform animal experiments, we did not initially include a PDX model. We have now shown that MDM2 inhibition via milademetan (RAIN-32, both 100 and 50 mg/kg, RAIN-Therapeutics) significantly reduced tumor growth in a PDX model of ER-positive mutant-GATA3

(D335Gfs) breast cancer resistant to fulvestrant (**Fig. 5h-i**). The results of the animal model (PDX) have been now included in the manuscript as follows:

Page 11-12 paragraph 2 (Results)

We further tested the antitumor effect of MDM2 inhibition over time in a PDX model of an ER-positive breast cancer harboring the *GATA3* p.D335Gfs mutation (**Fig. 5h**). PDX models have been shown to be the most representative preclinical models for drug development in several cancer types^{29,30}, as these models maintain the biological characteristics of the donor tumours, and there is a high degree of correlation between clinical response in patients and response to the same agent in PDX models generated from these patients^{29,30}. NOD/SCID mice were inoculated subcutaneously with p.D335Gfs *GATA3*-mutant breast tumor chunks (2-3 mm) for tumor development (**Fig. 5h**). When the mean tumor size reached approximately 150 mm³, PDX-inoculated mice were randomized for treatment for 29 days with one of the following: vehicle, fulvestrant (200 mg/kg), RAIN-32 (50 mg/kg), RAIN-32 (100 mg/kg) or a combination of fulvestrant (200mg/kg) and RAIN-32 (100mg/kg). Notably, while PDX-inoculated mice did not respond to treatment with fulvestrant, both 50 and 100 mg/kg doses of RAIN-32 were highly effective in reducing, and almost inhibiting, tumor growth (**Fig. 5i**). Similarly, treatment with fulvestrant in combination with RAIN-32 significantly reduced tumor growth when compared to treatment with fulvestrant alone (**Fig. 5i**).

Our *ex vivo* and *in vivo* data derived from PDOs and PDX models further support the use of MDM2 inhibition in the treatment of *GATA3*-mutant ER-positive breast cancer patients, in particular as a combination with first-line treatment with anti-estrogen therapy, or as alternative for those patients developing resistant to fulvestrant.

Page 29 paragraph 3 (Methods)

Mouse Husbandry

All mouse experiments were approved by and performed in accordance with the guidelines and regulations of the Animal Ethics Committee of the Association for Assessment and Accreditation of Laboratory Animal Care International. NOD/SCID female mice of 6-8 weeks of age were housed under pathogen-free conditions in individually ventilated cage (IVC) systems at constant temperature and humidity at the animal facilities of Crown Bioscience, Inc.

Primary *GATA3*^{p.D335Gfs} Breast Cancer Xenotransplantation and RAIN-32 Treatment

Clinical characteristics and immunophenotypic features of the BR5496 tumor sample used to develop these xenograft models are reported in **Supplementary Table 4**. PDX tumor fragments of 2 to 3 mm in diameter were subcutaneously xenografted into NOD/SCID mice. Each mouse was inoculated subcutaneously at the right flank region with BR5496 tumor chunk for tumor development. 40 mice were enrolled in the study. The PDX-inoculated mice were selected and randomly categorized into vehicle, fulvestrant, RAIN-32 50 mg/kg or RAIN-32 100 mg/kg groups (eight mice per group) when the mean tumor size reached approximately 144 mm³. The treatments started when the mean tumor size reached approximately 150 mm³ and lasted for 29 days. Randomization was performed based on the “Matched distribution” method using the StudyDirectorTM software, version 3.1.399.19 randomized block design. Tumor volumes were measured two times per week in two dimensions using a caliper, and the volume was expressed in mm³ using the formula: $V = 0.5 a \times b^2$ where *a* and *b* are the long and short diameters of the tumor, respectively. The entire procedures of dosing as well as tumor and body weight measurement were conducted in a Laminar Flow Cabinet.

Page 19 (Fig. 5 legend)

Fig. 5: *GATA3* mutations predict response to MDM2 inhibitors in ER-positive breast cancer PDOs and PDX. ... (g) Percentage of viable cells upon treatment with different dosages of RAIN-32 in *GATA3*-wild-type (gray) or *GATA3*-mutant (blue) PDOs. (h) Schematic representation of the PDX model and drug treatment. (i) Tumor growth curve of *GATA3* p.D335fs PDXs (n=6-8 mice) treated for 29 days with vehicle, fulvestrant (200 mg/kg), RAIN-32 (50 and 100 mg/kg) alone, or RAIN-32 in combination with fulvestrant. Data are mean \pm SD, $n \geq 4$ replicates from two or more independent experiments (c,d,e,g). Statistical significance was determined by two-tailed unpaired Student's t-tests, and by ANOVA multiple comparison.

Page 34 (Supplementary Table 4 legend)

Supplementary Table 4: Clinical characteristics and immunophenotypic features of the BR5496 tumor sample.

Page 36 (Supplementary Fig. 5 legend)

Supplementary Fig. 5: *GATA3* mutations predict response to MDM2 inhibitors in ER-positive breast cancer PDOs and PDX. ... (g) Log-dose response curve of MI-733 and RAIN-32 in MCF-7 cells. (h) Percentage of viable cells upon treatment with different dosages of MI-733 in *GATA3* wild-type or *GATA3*-mutant (met) PDOs.

6) What cohorts were used to analyze the frequencies of *GATA3*/*TP53* or *GATA3*/*PIK3CA* mutually exclusivity? Please mention these details in the results section or figure.

Authors: We apologize for the omission. ER-positive breast cancer mutational data for the *GATA3*, *TP53*, *PIK3CA* and *PTEN* genes and copy number status for *PTEN* were derived from the TCGA PanCancer Atlas⁷ and the METABRIC dataset⁶. We have now included this information in the figure legends.

7) What protocol did the authors use to establish the PDO? I assume it was reference 68 from Hans Clevers group. However, reference 67 is mentioned? Is this a typo? Moreover, apart from ER expression via IHC, it would be great to understand that these organoids are indeed estrogen responsive. For instance, do ER target genes get down-regulated upon fulvestrant, or upregulated upon estrogen?

Authors: We thank the reviewer for the questions. We have indeed used Hans Clever's group's protocol for PDO generation³¹. We apologize for the incorrect references, the correct manuscript is now referenced in the **Methods** section. While we did not test estrogen responsiveness directly on the PDOs, published *in vivo* data derived on these specific PDXs¹⁸ indicate that all three breast tumors were estrogen responsive. Indeed, treatment with fulvestrant resulted in tumor growth inhibition in HBCx-139 (*GATA3* WT) (Figure 2c in¹⁸) and in the downregulation of ER target genes in HBCx-137 (*GATA3* mutant primary) (Figure 1g of¹⁸). As for HBCx-169 (*GATA3* mutant, primary), the PDX was also estrogen dependent as estrogen removal from drinking water associated with ovariectomy resulted in tumor shrinkage (data unpublished). We have updated the manuscript as follows:

Page 10 paragraph 2 (Results)

Importantly, all tumor specimens were carrying a wild-type *TP53* gene, retained ER α and GATA3 expression (**Fig. 5b**, **Supplementary Fig. 5b**), and have been previously shown to be estrogen responsive *in vivo*¹⁸.

8) The mechanism linking the synthetic lethality between GATA3/MDM2 via the PI3K/AKT/mTOR pathway seems a bit weak and seems more of a strong correlation.

Authors: We thank the reviewer for this comment. We believe that the PI3K/AKT/mTOR pathway is not the only mechanism through which synthetic lethality between GATA3 and MDM2 is working. Indeed, our differential gene expression analysis has shown additional pathways involved, such as MYC and E2F targets (**Figure 6b**). Exploring the role of these additional players, as well as unraveling the relationship between p53, MDM2 and the GATA3 transcriptional landscape in breast cancer will eventually allow us to have the full picture. However, and within the scope of the present manuscript, we do believe the AKT/mTOR signalling pathway plays at least a partial role, and in the revised version of the manuscript we have provided additional data in support of our hypothesis. Specifically, to functionally validate the role of *GATA3* mutations in activating the mTOR signalling, we have assessed S6 protein phosphorylation upon rescue (overexpression) of the *GATA3* wild-type (**Supplementary Figure 6f**) in *GATA3*-mutant MCF-7 cells. In support of our hypothesis, rescuing *GATA3* wild-type in MCF-7 cells caused a 30% reduction in S6 phosphorylation 72 hours post transfection (**Supplementary Figure 6f**). Additionally, while idasanutlin treatment reduced S6 phosphorylation in control MCF-7 cells compared to DMSO, it did not significantly alter the S6 phosphorylation levels in *GATA3* rescued cells compared to their DMSO counterpart (**Supplementary Figure 6g**). Taken together, our new data provide evidence that *GATA3* mutations activate the mTOR signalling cascade, and that the effect of idasanutlin on the same pathway is dependent on the presence of a *GATA3* mutant protein. We believe that these additional data provide stronger support that the mTOR signalling pathway is at least partially involved in the synthetic lethal interaction between *GATA3* and MDM2. We have amended the manuscript accordingly:

Page 13 paragraph 2 (Results)

To functionally validate our hypothesis, we assessed the phosphorylation level of the S6 protein upon rescue of wild-type *GATA3* in MCF-7 cells (**Supplementary Fig. 6f**). Indeed, rescuing wild-type *GATA3* resulted in a 30% reduction of phospho-S6 72 hours post-transfection (**Supplementary Fig. 6f**). Additionally, pharmacological inhibition of MDM2 in control cells resulted in a reduction in phospho-S6 compared to DMSO, but failed to significantly alter phospho-S6 level in wild-type *GATA3*-rescued cells compared to their vehicle counterpart (**Supplementary Fig. 6g**). These results suggest that *GATA3* mutations activate the mTOR signalling cascade, and that the effect of MDM2 inhibition on the mTOR pathway is dependent on the presence of a *GATA3*-mutant protein... Taken together, our results show that the synthetic lethality between *GATA3* and MDM2 acts at least partially via the PI3K-Akt-mTOR signaling pathway.

Page 23 paragraph 2 (Methods)

For gene overexpression, log-phase MCF-7 and BT-474 breast cancer cells were seeded in 6-well plates at approximately 60%-80% confluence and transfected with control (pLV[Exp]-EGFP:T2A:Puro-CMV>Luc2 (VB190320-1059xxv)), *GATA3* wild-type (pRP[Exp]-EGFP/Puro-CMV>h*GATA3*[NM_001002295.2] (VB201028-1114vqf)) or *GATA3* mutant (D335Gfs) (pRP[Exp]-EGFP/Puro-CMV>{h*GATA3**(c.1006dup)} (VB210314-1087tvn)) expression vectors using JetPrime buffer and reagent (PolyPlus #101000027 and #201000003, **Supplementary Table 3**). 8 hours after transfection, the antibiotic-free medium was replaced with a complete medium.

Page 37 (Supplementary Fig. 6 legend)

Supplementary Fig. 6: ... (f) Immunoblot showing mutant (D335Gfs) and wild-type GATA3, actin, total and phospho-S6 (mTOR signaling pathway activation) at 48 and 72h post control or GATA3 wild-type overexpression in MCF-7 cells. (g) Immunoblot showing mutant (D335Gfs) and wild-type GATA3, actin, total and phospho-S6 (mTOR signaling pathway activation) at 48 h post control or GATA3 wild-type overexpression, and 24 h post treatment with vehicle or idasanutlin (12.5 μ M) in MCF-7 cells.

Reviewer #3 (Remarks to the Author):

In this study, the authors identified a new synthetic lethal interaction in human breast cancer. The authors showed that in estrogen receptor (ER)-positive breast cancers, mutations of GATA3 renders tumors to be synthetic lethal to MDM2 inhibition. Using a novel algorithm recently developed by the authors and through analysis of the breast cancer cell line (n=22) data from the large-scale, deep RNAi screen project DRIVE, the authors identified MDM2 as the top gene whose knock-down significantly reduced cell viability in the two GATA3-mutant breast cancer cell lines. The authors then went on to perform extensive in vitro and in vivo investigations to demonstrate that while mutation of GATA3 genes more ER+ breast cancer cells to become more proliferative and more tumorigenic, inhibition of MDM2 effectively reduces cell viability of ER+ breast cancer cells with GATA3 mutation in vitro and tumor formation in vivo. The translational potential of the synthetic lethal interaction of GATA3 mutation and MDM2 inhibition is further demonstrated using patient-derived organoids. Overall, this study has discovered a new synthetic lethal interaction in ER+ human breast cancer and has immediate translational potential. The study in general is well done and the manuscript was clearly written. It is recommended for publication in Nature Communications with the following changes/revisions.

Authors: We would like to thank the reviewer for the positive assessment of the study.

First, while the authors have built a very convincing case that GATA3 mutation in ER+ human breast cancer cells indeed makes the cancer cells becoming more sensitive to MDM2 inhibitor idasanutlin (about 5-times more sensitive in different assays) in vitro, MDM2 inhibitor did not stop tumor formation in zebrafish model (Figure 3a-d). Instead, it merely reduced the number of tumors. One would argue if GATA3 mutation in ER+ human breast cancer cells is truly synthetic lethal with MDM2 inhibition, MDM2 inhibitors should be able to completely stop tumor formation. Additionally, MCF-7 cells harbor GATA3 mutation but are not very sensitive to MDM2 inhibition.

Authors: We thank the reviewer for the comment. We believe that, given the nature of the assay (transient GATA3 gene knock-down, rather than stable knock-out or knock-in), it is expected that at least a certain amount of tumor cells would retain GATA3 expression at the time of drug treatment, and therefore be resistant to idasanutlin and still be able to proliferate once inoculated in the zebrafish. The differences in the extent of gene silencing among different clones might also account for different sensitivities towards treatment. Additionally, the short term duration of the assay itself (5 days) may also explain the presence of tumor formation. Despite the limitations of the model, we still believe it is of significance that idasanutlin treatment reduced tumor formation (more fish inoculated with GATA3 silenced cells did not harbor any tumor at the end of the 5-day treatment compared to control), as a result of reduced proliferation of inoculated tumor cells (measured as fluorescence-positive cells). We also would like to point out that in the revised version of the manuscript we have complemented our

in-vivo models (CAM and zebrafish) with a canonical ER-positive *GATA3*-mutant breast cancer PDX model (**Fig. 5h**). Indeed, we show that RAIN-32 (another MDM2 inhibitor) is effective in reducing tumor growth in the PDX derived from ER-positive *GATA3*-mutant (D335Gfs, same truncating mutation found in MCF-7 cells) breast cancer resistant to fulvestrant (**Fig. 5i**). Notably, the higher dose of RAIN-32 almost completely abolished tumor growth. We have added the new results as follows:

Page 11-12 paragraph 1 (Results)

We further tested the antitumor effect of MDM2 inhibition over time in a PDX model of an ER-positive breast cancer harboring the *GATA3* p.D335Gfs mutation (**Fig. 5h**). PDX models have been shown to be the most representative preclinical models for drug development in several cancer types^{29,30}, as these models maintain the biological characteristics of the donor tumours, and there is a high degree of correlation between clinical response in patients and response to the same agent in PDX models generated from these patients^{29,30}. NOD/SCID mice were inoculated subcutaneously with p.D335Gfs *GATA3*-mutant breast tumor chunks (2-3 mm) for tumor development (**Fig. 5h**). When the mean tumor size reached approximately 150 mm³, PDX-inoculated mice were randomized for treatment for 29 days with one of the following: vehicle, fulvestrant (200 mg/kg), RAIN-32 (50 mg/kg), RAIN-32 (100 mg/kg) or a combination of fulvestrant (200mg/kg) and RAIN-32 (100mg/kg). Notably, while PDX-inoculated mice did not respond to treatment with fulvestrant, both 50 and 100 mg/kg doses of RAIN-32 were highly effective in reducing, and almost inhibiting, tumor growth (**Fig. 5i**). Similarly, treatment with fulvestrant in combination with RAIN-32 significantly reduced tumor growth when compared to treatment with fulvestrant alone (**Fig. 5i**).

Page 29 paragraph 2 (Methods)

Mouse Husbandry

All mouse experiments were approved by and performed in accordance with the guidelines and regulations of the Animal Ethics Committee of the Association for Assessment and Accreditation of Laboratory Animal Care International. NOD/SCID female mice of 6-8 weeks of age were housed under pathogen-free conditions in individually ventilated cage (IVC) systems at constant temperature and humidity at the animal facilities of Crown Bioscience, Inc.

Primary *GATA3*^{p.D335Gfs} Breast Cancer Xenotransplantation and RAIN-32 Treatment

Clinical characteristics and immunophenotypic features of the BR5496 tumor sample used to develop these xenograft models are reported in **Supplementary Table 4**. PDX tumor fragments of 2 to 3 mm in diameter were subcutaneously xenografted into NOD/SCID mice. Each mouse was inoculated subcutaneously at the right flank region with BR5496 tumor chunk for tumor development. 40 mice were enrolled in the study. The PDX-inoculated mice were selected and randomly categorized into vehicle, fulvestrant, RAIN-32 50 mg/kg or RAIN-32 100 mg/kg groups (eight mice per group) when the mean tumor size reached approximately 144 mm³. The treatments started when the mean tumor size reached approximately 150 mm³ and lasted for 29 days. Randomization was performed based on the “Matched distribution” method using the StudyDirectorTM software, version 3.1.399.19 randomized block design. Tumor volumes were measured two times per week in two dimensions using a caliper, and the volume was expressed in mm³ using the formula: $V = 0.5 a \times b^2$ where *a* and *b*

are the long and short diameters of the tumor, respectively. The entire procedures of dosing as well as tumor and body weight measurement were conducted in a Laminar Flow Cabinet.

Page 19 (Fig. 5 legend)

Fig. 5: *GATA3* mutations predict response to MDM2 inhibitors in ER-positive breast cancer PDOs and PDX. ... (g) Percentage of viable cells upon treatment with different dosages of RAIN-32 in *GATA3*-wild-type (gray) or *GATA3*-mutant (blue) PDOs. (h) Schematic representation of the PDX model and drug treatment. (i) Tumor growth curve of *GATA3* p.D335fs PDXs (n=6-8 mice) treated for 29 days with vehicle, fulvestrant (200 mg/kg), RAIN-32 (50 and 100 mg/kg) alone, or RAIN-32 in combination with fulvestrant. Data are mean \pm SD, $n \geq 4$ replicates from two or more independent experiments (c,d,e,g). Statistical significance was determined by two-tailed unpaired Student's t-tests, and by ANOVA multiple comparison.

Page 34 (Supplementary Table 4 legend)

Supplementary Table 4: Clinical characteristics and immunophenotypic features of the BR5496 tumor sample.

Page 36 (Supplementary Fig. 5 legend)

Supplementary Fig. 5: *GATA3* mutations predict response to MDM2 inhibitors in ER-positive breast cancer PDOs and PDX. ... (g) Log-dose response curve of MI-733 and RAIN-32 in MCF-7 cells. (h) Percentage of viable cells upon treatment with different dosages of MI-733 in *GATA3* wild-type or *GATA3*-mutant (met) PDOs.

While most of the experiments were well designed, the use of very high concentrations of idasanutlin (~25 μ M) in some of the assays is problematic for data interpretation. At concentrations over 10 μ M, idasanutlin probably would have some off-target effects, in addition to MDM2 inhibition. Idasanutlin has a binding affinity in low nanomolar concentrations, there is no reason to use such high concentrations for the purpose of MDM2 inhibition. In addition, there are other clinical-grade MDM2 inhibitors (e.g. SAR405838 (MI-773), which are available commercially and could be used to further confirm that the effects observed for Idasanutlin were through MDM2 inhibition, instead of off-target activities.

Authors: We thank the reviewer for the important comment. When performing functional assays at 24 and 48 h post-treatment, a similar dose range of idasanutlin (5-10 μ M, with most of our assays on cell lines being performed with 12.5 μ M dose) has been previously reported in literature^{32,33}. Additionally, while the idasanutlin concentrations used in some assays (e.g. the 25 μ M used to pre-treat BT-474 cells before inoculation in the CAM and Zebrafish embryos) might look high at a first glance, especially considering the published potency, we would like to bring the attention to two aspects: 1) the reported IC50s are generally derived from 5-day treatment assays³², while we assessed cell viability at 72 h (for IC50 curve) and earlier (12 to 48 h post-treatment for most of the assays); when measuring cell viability upon 5-days treatment (PDOs treatment), we found *GATA3*-mutant PDOs to be responsive to idasanutlin at lower dose compared to MCF-7 (IC50=1.2 μ M, **Fig. 5d**). The latter indicates that the difference in treatment length might account for the difference between the reported potency and that obtained in MCF7 cells. Additionally, considering the relevance of PDOs in being more directly translatable to patients, indicate that *GATA3*-mutant tumors are indeed sensitive to idasanutlin. We would also like to underline that the selectivity of our idasanutlin treatment results was corroborated by our results obtained from selective MDM2 inhibition by siRNA. Furthermore, we

have added new results that showed that rescuing GATA3 wild-type expression in MCF-7 rescues the effect of the same dose of idasanutlin.

Following the reviewer's suggestion, we have validated our results using two additional MDM2 small molecule inhibitors, specifically MI-773 and RAIN-32 (**Fig. 5g-i** and **Supplementary Fig. 5g-h**). Both inhibitors showed higher potency in MCF-7 cells and breast cancer PDOs compared to idasanutlin (IC50 of 0.7 μ M and 1.7 μ M for RAIN-32 and MI-773, respectively. **Supplementary Fig. 5g**). As shown for idasanutlin, GATA3-mutant PDOs were significantly more sensitive to both RAIN-32 (**Fig. 5g**) and MI-773 (**Supplementary Fig. 5h**) at different dose levels. RAIN-32 additionally stopped tumor growth in a ER-positive GATA3-mutant breast cancer PDX model (**Fig. 5h-i**). Taken together, the original and the additional data provided in the revised version of the manuscript provide compelling evidence that the synthetic lethality between GATA3 and MDM2 is a specific effect, and not simply the result of some unspecific binding affinity of idasanutlin.

Page 11 paragraph 2 (Results)

To confirm that the effect of idasanutlin was indeed due to MDM2 inhibition rather than off-target effects, we tested additional clinical-grade MDM2 inhibitors, specifically MI-773 (SAR405838)³⁴ and RAIN-32 (milademetan³⁵, **Supplementary Fig. 5g**). Similar to idasanutlin, both inhibitors showed significantly higher efficacy in GATA3-mutant compared to wild-type PDOs (**Fig. 5g** and **Supplementary Fig. 5h**). We further tested the antitumor effect of MDM2 inhibition over time in a PDX model of an ER-positive breast cancer harboring the GATA3 p.D335Gfs mutation (**Fig. 5h**)... Notably, while PDX-inoculated mice did not respond to treatment with fulvestrant, both 50 and 100 mg/kg doses of RAIN-32 were highly effective in reducing, and almost inhibiting, tumor growth (**Fig. 5i**).

Page 19 (Fig. 5 legend)

Fig. 5: GATA3 mutations predict response to MDM2 inhibitors in ER-positive breast cancer PDOs and PDX. ... (g) Percentage of viable cells upon treatment with different dosages of RAIN-32 in GATA3-wild-type (gray) or GATA3-mutant (blue) PDOs. (h) Schematic representation of the PDX model and drug treatment. (i) Tumor growth curve of GATA3 p.D335fs PDXs (n=6-8 mice) treated for 29 days with vehicle, fulvestrant (200 mg/kg), RAIN-32 (50 and 100 mg/kg) alone, or RAIN-32 in combination with fulvestrant. Data are mean \pm SD, n \geq 4 replicates from two or more independent experiments (c,d,e,g). Statistical significance was determined by two-tailed unpaired Student's t-tests, and by ANOVA multiple comparison.

Page 24 (Methods)

10x10³ exponentially growing cells were plated in a 96-well plate. After 24 h, cells were treated with serial dilution of RG7388-idasanutlin, RAIN-32, MI-773 (**Supplementary Table 3**) or dimethyl sulfoxide (DMSO). DMSO served as the drug vehicle, and its final concentration was no more than 0.1%. Cell viability was measured after 72 h using CellTiter-Glo Luminescent Cell Viability Assay reagent. Results were normalized to the vehicle (DMSO).

Page 36 (Supplementary Fig. 5 legend)

Supplementary Fig. 5: GATA3 mutations predict response to MDM2 inhibitors in ER-positive breast cancer PDOs and PDX. ... (g) Log-dose response curve of MI-773 and RAIN-32 in MCF-7 cells. (h) Percentage of viable cells upon treatment with different dosages of MI-773 in GATA3 wild-

type or *GATA3*-mutant (met) PDOs.

References

1. McDonald, E. R., 3rd *et al.* Project DRIVE: A Compendium of Cancer Dependencies and Synthetic Lethal Relationships Uncovered by Large-Scale, Deep RNAi Screening. *Cell* **170**, 577–592.e10 (2017).
2. Srivatsa, S. *et al.* Discovery of synthetic lethal interactions from large-scale pan-cancer perturbation screens. doi:10.1101/810374.
3. Emmanuel, N. *et al.* Mutant *GATA3* Actively Promotes the Growth of Normal and Malignant Mammary Cells. *Anticancer Res.* **38**, 4435–4441 (2018).
4. Adomas, A. B. *et al.* Breast tumor specific mutation in *GATA3* affects physiological mechanisms regulating transcription factor turnover. *BMC Cancer* **14**, 278 (2014).
5. Barretina, J. *et al.* The Cancer Cell Line Encyclopedia enables predictive modelling of anticancer drug sensitivity. *Nature* **483**, 603–607 (2012).
6. Pereira, B. *et al.* The somatic mutation profiles of 2,433 breast cancers refines their genomic and transcriptomic landscapes. *Nat. Commun.* **7**, 11479 (2016).
7. Hoadley, K. A. *et al.* Cell-of-Origin Patterns Dominate the Molecular Classification of 10,000 Tumors from 33 Types of Cancer. *Cell* **173**, 291–304.e6 (2018).
8. Tichy, A. *et al.* The first in vivo multiparametric comparison of different radiation exposure

- biomarkers in human blood. *PLoS One* **13**, e0193412 (2018).
9. Higgins, B. *et al.* Preclinical optimization of MDM2 antagonist scheduling for cancer treatment by using a model-based approach. *Clin. Cancer Res.* **20**, 3742–3752 (2014).
 10. Haldi, M., Ton, C., Seng, W. L. & McGrath, P. Human melanoma cells transplanted into zebrafish proliferate, migrate, produce melanin, form masses and stimulate angiogenesis in zebrafish. *Angiogenesis* **9**, 139–151 (2006).
 11. Choi, J. *et al.* FoxH1 negatively modulates flk1 gene expression and vascular formation in zebrafish. *Developmental Biology* vol. 304 735–744 (2007).
 12. Dorsett, Y. & Tuschl, T. siRNAs: applications in functional genomics and potential as therapeutics. *Nat. Rev. Drug Discov.* **3**, 318–329 (2004).
 13. Han, H. RNA Interference to Knock Down Gene Expression. *Methods in Molecular Biology* 293–302 (2018) doi:10.1007/978-1-4939-7471-9_16.
 14. Svoboda, O. *et al.* Ex vivo tools for the clonal analysis of zebrafish hematopoiesis. *Nat. Protoc.* **11**, 1007–1020 (2016).
 15. Carapito, R. *et al.* Mutations in signal recognition particle SRP54 cause syndromic neutropenia with Shwachman-Diamond-like features. *J. Clin. Invest.* **127**, 4090–4103 (2017).
 16. Gaynor, K. U. *et al.* GATA3 mutations found in breast cancers may be associated with aberrant nuclear localization, reduced transactivation and cell invasiveness. *Horm. Cancer* **4**, 123–139 (2013).
 17. Mair, B. *et al.* Gain- and Loss-of-Function Mutations in the Breast Cancer Gene GATA3 Result in Differential Drug Sensitivity. *PLoS Genet.* **12**, e1006279 (2016).
 18. Montaudon, E. *et al.* PLK1 inhibition exhibits strong anti-tumoral activity in CCND1-driven breast cancer metastases with acquired palbociclib resistance. *Nat. Commun.* **11**, 4053 (2020).
 19. Bates, D. L., Chen, Y., Kim, G., Guo, L. & Chen, L. Crystal structures of multiple GATA zinc fingers bound to DNA reveal new insights into DNA recognition and self-association by GATA. *J. Mol. Biol.* **381**, 1292–1306 (2008).
 20. Cohen, H. *et al.* Shift in GATA3 functions, and GATA3 mutations, control progression and clinical presentation in breast cancer. *Breast Cancer Res.* **16**, 464 (2014).

21. Okawa, T. *et al.* A novel loss-of-function mutation of GATA3 (p.R299Q) in a Japanese family with Hypoparathyroidism, Deafness, and Renal Dysplasia (HDR) syndrome. *BMC Endocrine Disorders* vol. 15 (2015).
22. Kumar, R. *et al.* Structural and conformational changes induced by missense variants in the zinc finger domains of GATA3 involved in breast cancer. *RSC Advances* vol. 10 39640–39653 (2020).
23. Takaku, M. *et al.* GATA3 zinc finger 2 mutations reprogram the breast cancer transcriptional network. *Nat. Commun.* **9**, 1059 (2018).
24. Bertucci, F. *et al.* Genomic characterization of metastatic breast cancers. *Nature* **569**, 560–564 (2019).
25. Chakravarty, D. *et al.* OncoKB: A Precision Oncology Knowledge Base. *JCO Precis Oncol* **2017**, (2017).
26. Bahreini, A. *et al.* Mutation site and context dependent effects of ESR1 mutation in genome-edited breast cancer cell models. *Breast Cancer Res.* **19**, 60 (2017).
27. Kuang, Y. *et al.* Unraveling the clinicopathological features driving the emergence of ESR1 mutations in metastatic breast cancer. *npj Breast Cancer* vol. 4 (2018).
28. Jeselsohn, R. *et al.* Allele-Specific Chromatin Recruitment and Therapeutic Vulnerabilities of ESR1 Activating Mutations. *Cancer Cell* **33**, 173–186.e5 (2018).
29. Marangoni, E. & Poupon, M.-F. Patient-derived tumour xenografts as models for breast cancer drug development. *Current Opinion in Oncology* vol. 26 556–561 (2014).
30. Tentler, J. J. *et al.* Patient-derived tumour xenografts as models for oncology drug development. *Nat. Rev. Clin. Oncol.* **9**, 338–350 (2012).
31. Sachs, N. *et al.* A Living Biobank of Breast Cancer Organoids Captures Disease Heterogeneity. *Cell* **172**, 373–386.e10 (2018).
32. Ding, Q. *et al.* Discovery of RG7388, a Potent and Selective p53–MDM2 Inhibitor in Clinical Development. *Journal of Medicinal Chemistry* vol. 56 5979–5983 (2013).
33. Isermann, T. *et al.* Suppression of HSF1 activity by wildtype p53 creates a driving force for p53 loss-of-heterozygosity. *Nat. Commun.* **12**, 4019 (2021).

34. Wang, S. *et al.* SAR405838: an optimized inhibitor of MDM2-p53 interaction that induces complete and durable tumor regression. *Cancer Res.* **74**, 5855–5865 (2014).
35. Ishizawa, J. *et al.* Predictive Gene Signatures Determine Tumor Sensitivity to MDM2 Inhibition. *Cancer Res.* **78**, 2721–2731 (2018).

REVIEWERS' COMMENTS:

Reviewer #1 (Remarks to the Author):

All of my concerns have been addressed.

Reviewer #2 (Remarks to the Author):

The authors have addressed all my comments with further data and analyses and the manuscript has improved significantly.

Below you can find a point-by-point response to the comments:

REVIEWERS' COMMENTS:

Reviewer #1 (Remarks to the Author):

All of my concerns have been addressed.

Authors: we thank the reviewer for the comment

Reviewer #2 (Remarks to the Author):

The authors have addressed all my comments with further data and analyses and the manuscript has improved significantly.

Authors: we thank the reviewer for the comment